# ENABLING FINE-GRAINED OPERATING POINTS FOR BLACK-BOX LLMS

## ABSTRACT

Black-box Large Language Models (LLMs) provide practical and accessible alternatives to other machine learning methods, as they require minimal labeled data and machine learning expertise to develop solutions for various decision making problems. However, for applications that need operating with constraints on specific metrics (e.g., precision $\geq 95\%$), decision making with black-box LLMs remains unfavorable, due to their low numerical output cardinalities. This results in limited control over their operating points, preventing fine-grained adjustment of their decision making behavior. In this paper, we study using black-box LLMs as classifiers, focusing on efficiently improving their *operational granularity* without performance loss. Specifically, we first investigate the reasons behind their low-cardinality numerical outputs and show that they are biased towards generating *rounded* but informative verbalized probabilities. Then, we experiment with standard prompt engineering, uncertainty estimation and confidence elicitation techniques, and observe that they do not effectively improve operational granularity without sacrificing performance or increasing inference cost. Finally, we propose efficient approaches to significantly increase the number and diversity of available operating points. Our proposed approaches provide finer-grained operating points and achieve comparable to or better performance than the benchmark methods across 11 datasets and 3 LLMs.

## 1 INTRODUCTION

Black-box LLMs, accessible through cloud APIs, offer ready-to-use solutions for many real-world problems that previously required traditional machine learning approaches Ajwani et al. (2024); Fang et al. (2024). These models can be deployed for decision making tasks with minimal requirements for data annotation, computing infrastructure, or machine learning expertise. This accessibility has fueled their widespread adoption in various applications such as fraud detection, product classification, and medical diagnosis Min et al. (2021); Zeng et al. (2024).

Despite their versatility, these black-box models face limitations when applied to decision making tasks with operational or mission-critical constraints. Often, these constraints aim to adapt the mode of operation with respect to the expected risk associated with making a decision Davis & Goadrich (2006); Liu et al. (2017); Zadrozny (2004). One such example of an operational constraint is when the probability of making an incorrect decision is high; in such cases, a system may choose to delegate the decision to a larger model or a human-in-the-loop. Consequently, having finer-grained control over these operational metrics results in better optimization of the system's behavior. Typically, with predictive models that generate diverse and high-cardinality output distributions, traditional calibration techniques are used to conduct such behavior optimization on a Receiver Operating Characteristics (ROC) or precision-recall (PR) curve Guo et al. (2017). However, these techniques cannot be directly applied to Black-box LLMs due to lack of access to such a high-cardinality classification score distribution.

In this paper, we work under the assumption of an LLM being served behind a black-box service, where the user cannot access the LLM's weights, activations, logits, or token probabilities. We also assume that the LLM cannot be fine-tuned and is used as a frozen inference-only model. Many real-world widely-adopted LLM services fall under this business model. Then, in a binary classification setting, one immediate way to make this black-box LLM emulate a *white-box* machine learning

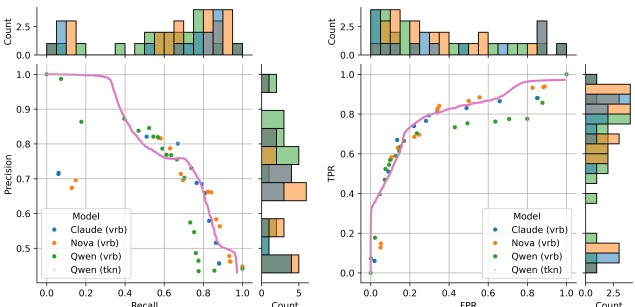

Figure 1: PR and ROC curves of 3 LLMs' verbalized probability estimates (vrb) together with Qwen's decision token probabilities (tkn) across 11 datasets. Histograms of verbalized probabilities over individual axes are also provided to visualize the sparsity along these axes. All three LLMs' verbalized probability distributions imply low cardinality and diversity of operating points, while using token probabilities provides a highly continuous curve (its histogram is omitted for brevity).

model is to prompt the LLM to output a verbalized quantity associated with its decision Lin et al. (2022); Tian et al. (2023); Xiong et al. (2023); Wen et al. (2024). When prompted, this score can be generated as the numerical string representation of the probability as a float between 0 and 1, or indirectly converted from a grading scale (e.g., 1 to 5, A to Z, and very confident to not confident).

Let us take the example of direct probability-emulation output. In this setting, given a classification dataset $\{(\mathbf{x}_i, y_i)\}_{i=1}^N$, $y_i \in \{c_1, c_2, ..., c_n\}$, the black-box LLM is prompted to estimate the class probabilities $p(y_i = c_j | \mathbf{x}_i) \, \forall i, j$ and return them as part of its textual output. One issue we observe is that the distribution of returned probabilities contains a small number of distinct elements. Figure 1 compares the precision-recall (PR) curves of such probability-emulation of 3 LLMs, as well as the white-box LLM Qwen's DeepSeek-AI (2025) token probabilities across 11 datasets. Although directly prompted to generate continuous class probabilities, we observe that simple prompting has a significantly lower number of distinct predictions compared to using token probabilities that are not accessible in black-box setting. Consequently, number of unique model score thresholds that correspond to different operating points is also low, resulting in coarse *operational granularity*. Given an operational measure $\alpha$, operational granularity measures the smallest adjustment that can always be made to $\alpha$. This is problematic because many business-critical and mission-critical real-world applications rely on complete precision-recall (PR) or Receiver Operating Characteristic (ROC) curves to select the appropriate operating point for the system, based on specific operational requirements like high recall or high precision. However, with such a limited set of probabilities from the black-box LLM, it is not possible to finely adjust the system behavior to a desired operating point.

To address these limitations, we investigate the reasons behind the low-cardinality output distributions of LLMs, and we propose efficient approaches to map black-box LLMs' verbalized probabilities to a distribution of predictions that has fine-grained coverage of spaces described by two widely-used operational metrics: PR and ROC curves Flach & Kull (2015). These approaches involve introducing parameterized noise to LLM outputs, where the parameters are estimated from data. These approaches can be interpreted as using a known continuous noise distribution (e.g., Gaussian noise) as prior, which inherently has the cardinality of the continuum $|\mathbb{R}|$. Then, guided by black-box LLM outputs, we learn functions $f$ to transform this prior to a prediction distribution with fine *operational granularity*.

Our primary contributions include: (1) demonstrating the prevalence and impact of low cardinality in LLM verbalized scores through extensive experiments, (2) providing a hypothesis for this behavior with supporting experiments, and (3) offering simple yet effective solutions that outperform SOTA confidence elicitation and uncertainty estimation techniques on the task of increasing cardinality (precisely operational granularity) while maintaining predictive performance. Our focus is not on better model calibration or improving model performance in terms of task specific metrics but on improving granularity. To do so, we modify the LLM predicted scores using noise and do not alter the initial predictions by the LLM. We show through extensive experiments that it is not trivial to improve granularity with just better prompting techniques and that model fine-tuning might be necessary to fundamentally modify the LLM scores. This is outside the scope of our work.

## 2 RELATED WORK

We focus on reviewing uncertainty estimation and confidence elicitation studies, as they are the most relevant directions to our problem. In Appendix A.1, we also discuss works that study model calibration and numerical capabilities of LLMs.

**Black-Box Uncertainty Estimation.** In this paper, we aim to extract highly granular classification scores from black-box LLMs. One way to tackle this problem is to use uncertainty estimation techniques to calculate a continuous measure of uncertainty and combine it with the classification decision. Lin et al. (2024) proposes and compares a group of simple sampling-based uncertainty metrics that utilize the number of semantic sets, sum of eigenvalues of the graph Laplacian and the degree matrices formed based on the output distributions. Tsai et al. (2024) introduces an efficient approach that utilizes point-wise dependency neural estimation to assess decision trustworthiness for stepwise decision planning. Manakul et al. (2023) proposes a hallucination detection framework by aggregating multiple evaluations of stochastically-generated LLM responses. Wagner et al. (2024) creates a confusion matrix by injecting bias into prompts for each class and aggregates the measurements in the matrix to produce an uncertainty label. Although uncertainty estimation methods tend to increase the LLM output cardinality, they require tens to hundreds of repeated LLM calls per sample to converge and produce sufficiently continuous output distributions.

**Confidence Elicitation.** An efficient alternative to sampling-based uncertainty estimation is to use prompting strategies to extract self-assessments associated LLM decisions. Tian et al. (2023); Lin et al. (2022) find that verbalized confidences often align better with the actual correctness of the model's answers compared to internally calculated token probabilities and predictive uncertainty. Xiong et al. (2023) observes that LLMs tend to be overconfident with their elicitation, possibly mimicking the style of human confidence expressions. The authors report that the performance difference between using token probabilities of white-box models and using elicited confidences of black-box models is small, however, there is no single technique that consistently works the best. Notably, the authors also observe that the verbalized confidences of LLMs usually fall in the 80-100 range, and are expressed in multiples of 5. However, they do not further investigate this phenomena. Zhou et al. (2023b) studies the relationship between the epistemic markers injected into the prompts and the corresponding confidence expression verbalized by LLMs. They find that LLMs tend to be biased from the observed language use in prompts, rather than truly expressing their own epistemic uncertainty. These studies commonly observe that confidence elicitation produces an informative but potentially biased signal, and they propose prompting approaches to mitigate these biases. Also, these approaches do not significantly increase the cardinality or diversity of the LLM-generated confidences, and they do not work consistently across datasets.

## 3 METHODOLOGY

### 3.1 BACKGROUND

In real-world applications, due to the constraints and requirements of the use case, machine learning models often need to be evaluated by a combination of metrics. For example, a fraud detection model that declines banking activities may need to operate with a low false positive rate but a high recall to maintain good customer experience, while minimizing fraudulent activity. For such use cases, using one-dimensional performance metrics such as accuracy does not provide sufficient insight into model behavior. Instead, multi-dimensional performance summaries such as ROC and PR curves are utilized Davis & Goadrich (2006). As they are common choices for decision making problems, we focus on these two curves. We also limit the discussion to binary classification, however, our analyses and proposed approaches can be extended to multiclass setting by taking a 1-vs-rest or 1-vs-1 approach.

**Operational Curves.** In a binary decision problem with positive (P) and negative (N) classes, let $\mathbf{x}$ be an input instance with an unknown class label $y \in \{\mathrm{P}, \mathrm{N}\}$. Let a classifier estimate the probability of input belonging to class P: $\hat{y} = p(y = \mathrm{P}|\mathbf{x}) \in [0, 1]$. We can use a threshold $th \in [0, 1]$ to map the probability estimate to a decision P if $\hat{y} > th$, and to N otherwise. Then, a distribution of decisions over a set of instances can be summarized in a *confusion matrix* that contains counts of true positives (TP), false positives (FP), true negatives (TN) and false negatives (FN). Notably, a confusion matrix can be seen as a function of a multiset of prediction-label pairs $[(y, \hat{y})_i]_{i=1}^{N}$ and a threshold $th_j$, since the distribution of true and false decisions change based on these variables.

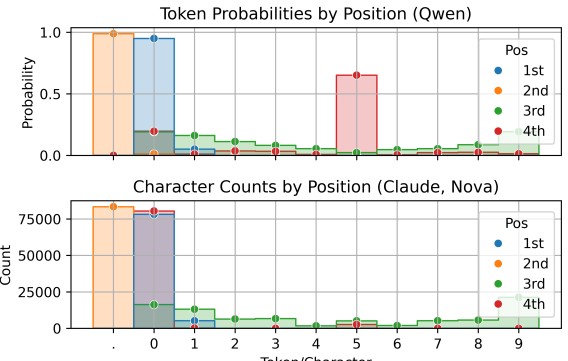

| Model | $|\hat{y}| \uparrow$ | $g^{\mathrm{pre}} \downarrow$ | $g^{\mathrm{rec}} \downarrow$ | $g^{\mathrm{fpr}} \downarrow$ |
|---|---|---|---|---|
| Claude (vrb) | 16 | 0.080 | 0.450 | 0.191 |
| Nova (vrb) | 23 | 0.085 | 0.424 | 0.270 |
| Qwen (vrb) | 21 | 0.128 | 0.219 | 0.194 |
| Qwen (tkn) | 41671 | 0.001 | 0.028 | 0.028 |

Figure 2 (b): Output cardinalities ($|\hat{y}|$) and operational granularities ($g$) of three LLMs across 11 datasets (granularities are truncated to the third decimal points). We observe that (1) verbalized probabilities produce orders of magnitude coarser-grained operational curves and lower-cardinality output distributions than using token probabilities, and (2) higher cardinality does not always translate to finer-granularity as many unique predictions may densely populate a sub-region and leave the rest of the space sparse.

Figure 2 (a): All 3 LLMs are heavily biased towards verbalizing probability estimates that end with `"0"` and `"5"`s. On the other hand, the distributions of the 3rd characters are spread across all tokens (digits) with slights bias towards extremes. Interestingly we observe that even though all three models' top 2 most frequent tokens in position 4 are consistent, Qwen's highest probability token is `"5"` while the others' is `"0"`.

Specifically, given this set of prediction and label pairs, let $M_{th_j} = \begin{bmatrix} \mathrm{TP}_{th_j} & \mathrm{TN}_{th_j} \\ \mathrm{FP}_{th_j} & \mathrm{FN}_{th_j} \end{bmatrix}$ denote the confusion matrix constructed using the threshold $th_j$. $M_{th_j}$ can be used to construct a point in either ROC space $\mathbb{R}^{fpr} \times \mathbb{R}^{tpr}$ as $\mathbf{x}_j^{ROC} : \left( \dfrac{\mathrm{FP}_{th_j}}{\mathrm{FP}_{th_j} + \mathrm{TP}_{th_j}}, \dfrac{\mathrm{TP}_{th_j}}{\mathrm{FN}_{th_j} + \mathrm{TP}_{th_j}} \right)$, or a point in the PR space $\mathbb{R}^{rec} \times \mathbb{R}^{pre}$ as $\mathbf{x}_j^{PR} : \left( \dfrac{\mathrm{TP}_{th_j}}{\mathrm{FN}_{th_j} + \mathrm{TP}_{th_j}}, \dfrac{\mathrm{TP}_{th_j}}{\mathrm{FP}_{th_j} + \mathrm{TP}_{th_j}} \right)$, where $fpr$ and $tpr$ denote false and true positive rates, and $rec$ and $pre$ denote recall and precision, respectively. Then, corresponding ROC and PR curves can be expressed as the following sets:

$$\mathrm{ROC}\big([(y,\hat{y})_i]_{i=1}^N\big) = \big\{ \mathbf{x}_j^{ROC} \,\forall\, th_j \in \{\hat{y}_i\} \big\}, \quad \mathrm{PR}\big([(y,\hat{y})_i]_{i=1}^N\big) = \big\{ \mathbf{x}_j^{PR} \,\forall\, th_j \in \{\hat{y}_i\} \big\}. \quad (1)$$

**Operational Granularity.** Both ROC and PR curves describe mappings from multisets to sets, because the thresholds to produce points in these spaces are defined in the set of unique model predictions $\{\hat{y}_i\}$. The distributions of the points in these sets directly express the operational behavior of the model that generates them. Depending on how sensitive the use case is to the operational metrics that these spaces consist of, the number of points to sufficiently cover the full operational range may vary. For example, being able to control the precision of a model in $0.25$ increments may be sufficient for sentiment classification, but not for medical diagnosis. Given a set of points $S \in \mathbb{R}^a \times \mathbb{R}^b$ that describe a model's operational behavior (e.g., $\mathrm{ROC} \in \mathbb{R}^{fpr} \times \mathbb{R}^{tpr}$), we define the *operational granularity* of $S$ along an axis $a$ as

$$g^a(S) = \operatorname*{argmin}_{s \geq 0} \left\{ s \mid \forall m \in \mathbb{Z} \cap \big[0, \lceil 1/s \rceil\,\big], \exists p_i \in \Pi_a(S) : m \cdot s \leq p_i < (m+1) \cdot s \right\}, \quad (2)$$

where $\Pi_a(S)$ denotes the projection of points in $S$ onto $a$. Intuitively, $g^a(S)$ is the cell size of the minimum-width uniform grid that $\Pi_a(S)$ covers, capturing the granularity of adjustment that can be consistently done along the range in which the operational metric $a$ is defined. Then, a lower value of $g^a(S)$ would imply a consistently finer-grained control over the metric $a$.

### 3.2 Analysis of LLM-Generated Probability Estimates

In this section, we investigate the verbalized probability estimates $\hat{y}^{\mathrm{vrb}}$ generated by three LLMs: the black-box `Anthropic Claude 3.0 Sonnet` (Claude) Anthropic (2024) and `Amazon Nova Pro 1.0` (Nova) Amazon (2025b), and the white-box `DeepSeek R1 Distill-Qwen-32B` (Qwen) DeepSeek-AI (2025). We also investigate Qwen's token probability distribution $\hat{y}^{\mathrm{tkn}}$ corresponding to the decision token to gain further insights. For this analysis, we combine samples from 11 binary classification datasets (see Appendix A.2) and prompt the LLMs to verbalize numerical

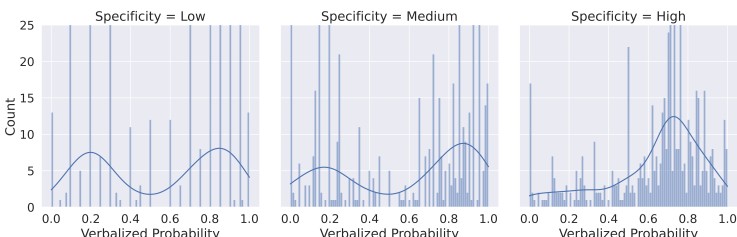

Figure 3: Verbalized probability distributions of Qwen using prompts with low, medium and high specificity. The output cardinality can be increased by providing additional instructions on how to conduct the task.

classification scores given input instances (see Appendix A.3 for the prompts used). We analyze their score distribution cardinalities, PR and ROC curves, and operational granularities using equation 2.

**Verbalized classification scores have low cardinality.** Figure 1 depicts the PR and ROC curves of the distributions $\hat{y}^{\text{vrb}}_{\text{Claude}}$, $\hat{y}^{\text{vrb}}_{\text{Nova}}$, $\hat{y}^{\text{vrb}}_{\text{Qwen}}$ and when prompted to output class probabilities, and $\hat{y}^{\text{tkn}}_{\text{Qwen}}$ when prompted to output binary decisions (e.g., *yes* or *no*). From the figure, it can be observed that all three LLMs generate sparsely populated PR and ROC curves when verbalized probabilities are used, while the token probabilities of Qwen (i.e., $\hat{y}^{tkn}_{\text{Qwen}}$) form continuous curves. Figure 2 (b): shows output cardinalities and operational granularities of these four variants. We observe that using verbalized probabilities only provide 16-23 unique values, while token probabilities provide 41671 unique values. Notably, we observe minimal change in cardinality when we prompt the LLMs to generate scores in [0, 100], generate and combine a coarse and a fine grained score, generate 20 scores within a single output and conduct post-processing to aggregate them, and generate output by using a context of 20 example scores in desired high-precision form. See Appendix A.4 for experiment results with these variants. We additionally consider two prompt based approaches from state-of-the-art confidence estimation works Tian et al. (2023); Xiong et al. (2023). These are two stage approaches (requiring 2 LLM calls per instance) with the LLM first predicting just the class label and then provided with this label, predicting the likelihood of its correctness. One of these variants uses chain-of-thought prompting in addition to the two-stage approach.

**Verbalized probability generation is biased towards round but informative numbers.** In order to gain additional insights into low-cardinality generations, we dive deeper into token probabilities associated with the generations $\hat{y}^{\text{vrb}}_{\text{Qwen}}$ character by character. Note that, this is different than $\hat{y}^{\text{tkn}}_{\text{Qwen}}$ that corresponds to the token probabilities associated to binary decisions. Specifically, given a verbalized probability $\hat{y}^{\text{vrb}}[i]$, we individually analyze the token probability distribution of each character by their position. For example, let $\hat{y}^{\text{vrb}}_1 = $"0.95". Then, the positions 1 and 3 would correspond to the tokens "0" and "9", respectively. For the black-box Claude and Nova, because the token probabilities are not available, we conduct the same analysis by counting the characters occurring at each position. Figure 2 (a): depicts the distribution of the token probabilities and character counts by position across 11 datasets. From the figure, it can be observed that when simply asked to verbalize class probabilities, all three LLMs chose to end their generations with "0" and "5" tokens most of the time, regardless of the task. We hypothesize that this is due to the fact that LLMs are trained and aligned by heavily relying on human-generated data, and humans tend to prefer expressing quantities in round numbers as their cognitive bias Pope & Simonsohn (2011); Zhou et al. (2023b). When this bias is reflected to LLM generations, it creates a many-to-one mapping from an internal high-precision representation (e.g., discrete token probabilities) to a round number, reducing the cardinality of generations. We refer to this generation behavior of LLMs as *rounding bias*. To be clear, in this context, we are using the definition of roundness as described by Ferson et al. (2015): a number ending in the digit 5 is considered rounder than any number ending in 1, 2, 3, 4, 6, 7, 8, or 9. However, a number ending in 5 is considered less round than any number ending in 0. From the figure, we also observe that the distribution in the third position is spread across all numerical tokens, while having slight bias towards extremes. Notably, even though the verbalized probabilities are heavily impacted by the rounding bias, they still imply operational curves that are visibly correlated with the ones generated using $\hat{y}^{\text{tkn}}_{\text{Qwen}}$ as shown in Figure 1.

**Increased specificity in prompt can reduce rounding bias.** We aim to understand why verbalized probabilities are heavily impacted by rounding bias, while LLMs can conduct operations such as

| Prompt Specificity | $|\hat{y}| \uparrow$ | $g^{\text{pre}} \downarrow$ | $g^{\text{rec}} \downarrow$ | $g^{\text{fpr}} \downarrow$ | PRAUC↑ | AUROC↑ |
|---|---|---|---|---|---|---|
| Low | 28 | 0.091 | 0.188 | 0.297 | 0.775 | 0.792 |
| Medium | 89 | 0.031 | 0.117 | 0.104 | 0.713 | 0.748 |
| High | 179 | 0.010 | 0.050 | 0.042 | 0.633 | 0.663 |

Table 1: Cardinality, granularity and AUCs of Qwen's verbalized probability distributions when prompted with Low, Medium and High specificity. As the specificity increases, cardinalities and granularities improve. However, the increased specificity may impact the performance.

simple algebra with reasonable precision. We hypothesize that tasks with high ambiguity leave room for the rounding bias to impact the output, as the steps of conducting the task are not explicit. We design an experiment using prompts with increasing levels of specificity to help reduce task ambiguity. The low specificity prompt (A.3.2) asks the model to directly output a class probability value, the medium specificity prompt (A.3.3) instructs the model to first extract numerical features and then combine them, and the high specificity prompt (A.3.4) additionally instructs the model to combine the extracted features using a parameterized function with weights assigned using the LLM's own sense of feature importance. Figure 3 and Table 1 depict the histograms of verbalized probability distributions as well as cardinality, operational granularity and AUC measurements using these three prompts. We find that as the specificity increases, the granularity and cardinality of the output improves. However, increased specificity may impact the performance. Although, with further trial and error, it may be possible to maintain the performance, these results show that the impact of increased specificity to performance is not always predictable. We provide additional ablations with these prompts in Appendix A.5.

## 3.3 PROPOSED APPROACHES

In Section 3.2, we observe that LLM-generated verbalized probabilities have rounding bias and attempting to resolve it with prompting is non-trivial as it may negatively impact performance. Additionally, we see that verbalized probabilities imply PR and ROC curves that still behave similarly to using token probabilities. Motivated by these observations, we propose approaches to enrich verbalized probabilities with noise introduced in various ways. As the cardinality of noise sampled from a continuous distribution would approach to infinity with increasing sample size, we expect it to help introduce smoothness and diversity to the verbalized probability distributions, if it is aggregated with the original probabilities *carefully*.

Given a dataset $D = \{(\mathbf{x}_i, y_i)\}_{i=1}^{N}$ where $y_i \in \{0, 1\}$, let a black-box LLM output the verbalized probability $\hat{y}_i^{\text{vrb}}$ that estimates $p(y_i = 1|\mathbf{x}_i)$. Then, our objective can be *loosely* formulated as a constrained optimization problem:

$$\underset{\theta_f}{\arg\min} \left[ g^a(S_f), g^b(S_f) \right] \quad \text{s.t.} \quad \sum_{i=1}^{N} \ell(y_i, \hat{y}_i = f(\hat{y}_i^{\text{vrb}}; \theta_f)) \leq \sum_{i=1}^{N} \ell(y_i, \hat{y}_i^{\text{vrb}}), \quad (3)$$

where $S_f$ denotes the operational curve $S([y, f(\hat{y}_i; \theta_f)]_{i=1}^{N})$ defined for $S \in \{\text{ROC}, \text{PR}\}$ in Equation equation 1, $g(S_f)$ is the operational granularity of $S_f$ from Equation equation 2, $\hat{y} = f(.; \theta_f)$ is a function that maps verbalized probabilities $[\hat{y}_i^{\text{vrb}}]_{i=1}^{N}$ to an output distribution $[\hat{y}_i]_{i=1}^{N}$ that estimates class probabilities, and $\ell$ is a function that measures the classification loss. Then, the objective is to find a function $f$ (by estimating $\theta_f$) that minimizes the operational granularity for $a$ and $b$ of $S \in \mathbb{R}^{a \times b}$, while keeping the empirical risk bounded by the loss incurred by the original verbalized predictions $[\hat{y}_i^{\text{vrb}}]_{i=1}^{N}$. One way to attempt solving this optimization problem is to observe that according to Equations equation 1 and equation 2, $g^a(S_f)$ and $g^b(S_f)$ depend on the cardinality $\left|[\hat{y}_i]_{i=1}^{N}\right|$ and the entropy $H\left([\hat{y}_i]_{i=1}^{N}\right)$ of the output distribution. In fact, setting $\left|[\hat{y}_i]_{i=1}^{N}\right|$ to samples from the standard uniform distribution $U(0, 1)$ would minimize this objective, however, would also violate the inequality constraint. Based on this observation, instead of directly minimizing $\left[g^a(S_f), g^b(S_f)\right]$, we reformulate the objective of our first proposed variant as follows:

$$\max(w) \quad \text{s.t.} \quad \sum_{i=1}^{N} \ell(y_i, \hat{y}_i = \text{clip}_{[0,1]}(z_i w + \hat{y}_i^{\text{vrb}})) \leq \sum_{i=1}^{N} \ell(y_i, \hat{y}_i^{\text{vrb}}), \quad w > 0, \quad (4)$$

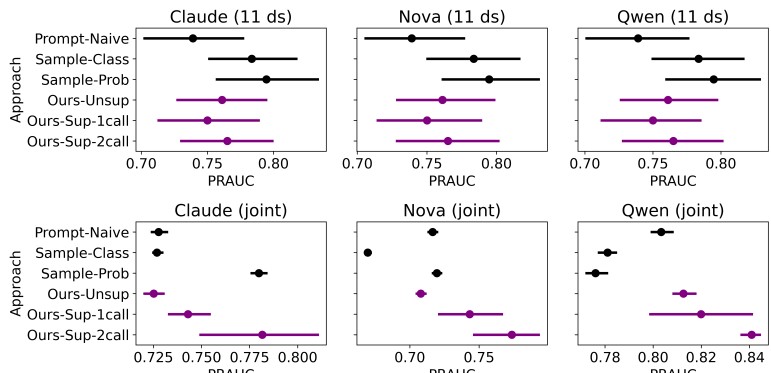

Figure 4: PRAUC of different approaches over 5 different train/test splits with $2\sigma$ error bars. Proposed methods are shown in purple. **(Top)** Aggregated results over 11 datasets. While naive prompting is usually worse than other methods, no method consistently outperforms the rest. Since a majority of these datasets are small and have nearly monotonic PR curves, the proposed supervised methods do not provide significant boosts. **(Bottom)** Results on combined dataset. Supervised approaches vastly outperform the rest including the sampling approaches requiring 20 LLM calls per instance.

where $\text{clip}_{[0,1]}(x)$ maps $x$ to the closest value in $[0,1]$, and $z \sim U(0,1)$. Note that in Equation equation 4, $w$ controls the amount of noise introduced to the verbalized predictions. Then, Equation equation 4 aims to find the largest $w$ that does not impact the classification performance. Note that, to maintain the classification performance in PR and ROC spaces, it is not required to directly calculate the classification losses $\ell$. As long as the relative ranks of the predictions remain the same, the performance in these spaces will remain unchanged. Then, given a list of verbalized probabilities $[\hat{y}_i^{\text{vrb}}]_{i=1}^n$, one can set the $w$ such that $z_{n+1}w + \hat{y}_{n+1}^{\text{vrb}} < \hat{y}_i^{\text{vrb}}$, where $\hat{y}_i^{\text{vrb}}$ is the smallest probability in the list that is larger than $\hat{y}_{n+1}^{\text{vrb}}$. We refer to this approach as `Ours-Unsup`.

We empirically observe that using `Ours-Unsup` produces operational curves by interpolating the curves implied by the verbalized probabilities. However, when these probabilities do not have a sufficiently monotonous relationship between their empirical performance, there may be room to correct this behavior and lift performance using ground truth. This is typically the case for machine learning models that operate in the zero-shot setting, as their score distributions are not calibrated to the corresponding task. In order to achieve this, we extend equation 4 to a supervised learning setting, reformulating the objective as finding a continuous distribution with high entropy and that can be transformed to an output distribution that minimizes the empirical loss:

$$\underset{\theta_f, w>0}{\arg\min} \sum_{i=1}^{N} \ell(y_i, \hat{y}_i = \text{sig}(\frac{z_i}{w} + f(\hat{y}_i^{\text{vrb}}; \theta_f))) + \lambda w, \tag{5}$$

where $z \sim \mathcal{N}(0,1)$ and $\text{sig}(x) = 1/(1 + e^{-x})$. In Equation equation 5, $f$ models additive corrections made to the samples from a Gaussian distribution with the standard deviation $\sigma \propto 1/w$. Since the differential entropy of a Gaussian (i.e., $\ln(\sigma\sqrt{2\pi e})$) is proportional to its standard deviation, decreasing the value of a positive $w$ would increase the entropy of the sample distribution that is corrected by $f$ and $[z_i/w]_{i=1}^N$. Consequently, $\lambda$ controls the magnitude of regularization that pushes the entropy up, and we leave it as a hyperparameter. $(w, \theta_f)$ is estimated using a ReLU network Hanin & Rolnick (2019) by setting $\ell$ to cross-entropy loss Mao et al. (2023). We refer to this approach as `Ours-Sup-1call`. It is important to note that Equation equation 5 learns a mapping from $(z_i, \hat{y}_i^{\text{vrb}})$ to $y_i$, without having visibility of input features $\mathbf{x}_i$. Therefore, when $w \to \infty$, the mapping $f$ reduces to a *calibrator* that learns to correct $\hat{y}^{\text{vrb}}$'s mistakes that is consistently repeated across different input instances. In Appendix A.7, we provide ablation studies to show that jointly learning to introduce noise and correcting miscalibrated behavior (i.e., equation 5) performs better than conducting these steps separately.

## 4 EXPERIMENTS

We empirically demonstrate the improvements in performance and granularity of operating points using the proposed approaches. We use 11 publicly available binary classification datasets to

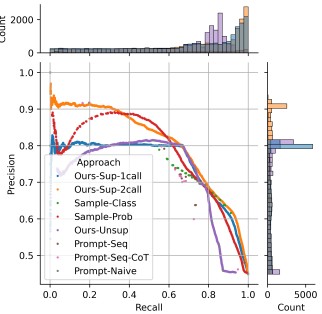

| Approach | Calls | $|\hat{y}|$ ↑ | $g^{pre}$ ↓ | $g^{rec}$ ↓ | $g^{fpr}$ ↓ | PRAUC ↑ |
|---|---|---|---|---|---|---|
| Prompt-Naive | 1 | 15 | 0.078 | 0.451 | 0.207 | 0.73(0.01) |
| Prompt-Seq | 2 | 16 | 0.157 | 0.377 | 0.525 | 0.72(0.01) |
| Prompt-Seq-CoT | 2 | 15 | 0.137 | 0.399 | 0.504 | 0.71(0.01) |
| Sample-Class | 20 | 97 | 0.168 | 0.552 | 0.124 | 0.73(0.00) |
| Sample-Prob | 20 | 2291 | **0.014** | 0.018 | 0.010 | **0.78**(0.01) |
| lightpurple Ours-Unsup | 1 | 18673 | 0.023 | 0.037 | 0.011 | 0.72(0.01) |
| lightpurple Ours-Sup-1call | 1 | 20596 | 0.016 | **0.001** | **0.004** | 0.74(0.02) |
| lightpurple Ours-Sup-2call | 2 | **20607** | 0.016 | **0.001** | 0.005 | **0.79**(0.05) |

Figure 5 (b): Operational granularity of the baselines and proposed approaches measured along precision ($pre$), recall ($rec$) and false positive rate ($fpr$) axes to cover both PR and ROC spaces, as well as their output cardinalities ($car$). Granularities are calculated using Equation equation 2. Proposed methods increase cardinality by nearly 3 orders of magnitude providing great granularity along all 3 axes. $2\sigma-$ error margins are denoted in parenthesis for PRAUC.

Figure 5 (a): Scatter plot of precision-recall values for all possible thresholds. The plots along the margin show the histogram of operating points for each method. The proposed methods have significantly higher output cardinality, providing better control over the operating point, while simultaneously improving the performance.

evaluate the effectiveness of the proposed approach, covering a variety of applications from sentiment classification to heart disease detection. Dataset sources, sizes and processing details can be found in Appendix A.2. Similar to Section 3.2, we experiment with three LLMs of varying parameter counts - `Anthropic Claude 3.0 Sonnet` (Claude), `Amazon Nova Pro 1.0` (Nova) and `DeepSeek R1 Distill-Qwen-32B` (Qwen). We implement `Ours-Sup-1call` as an MLP with 2 hidden-layers of width 8 and with ReLU activation function, and tune learning rates via grid search within $[0.01, 0.05, 0.1]$. We also conduct grid search to find a suitable value for $\lambda$ from Equation equation 5 within $\text{LogUniform}(1e^{-4}, 1e^{-1})$.

**Benchmarks.** `Prompt-Naive`, with a simple prompt to predict verbalized scores, serves as our primary baseline (Appendix A.3.1). Following Tian et al. (2023); Xiong et al. (2023), we consider a two-stage prompting approach where only a class prediction is obtained in the first LLM call and only the confidence for the predicted class is obtained in the second LLM call. We use both regular (termed `Prompt-Seq`) and chain-of-thought (termed `Prompt-Seq-CoT`) variants for this approach. Additionally, as described in Sec. 3.2, we consider several prompt methods for comparison. However, all these approaches fail to significantly improve cardinality and might result in performance drops. For brevity, we compare against the baseline and state-of-the-art prompt variants here and provide the remaining results in the Appendix. For Qwen, we directly use the token probabilities for the decision token as the classification score. In the absence of these probabilities for the black-box Claude and Nova models, we estimate them by setting the temperature to 1, repeatedly sampling outputs, and calculating the ratio of times the most frequent class was predicted. This method is termed as `Sample-Class`. It is straightforward to show that when temperature is set to 1, as the number of samples approaches infinity, the estimated probability approaches the token probability of the decision token. For each input instance, we sample black-box LLMs 20 times to conduct this approximation. `Sample-Prob` can be seen as a mix of verbalizing and sampling approaches where multiple samples per instance are obtained as in `Sample-Class` but the verbalized probabilities are predicted for each instance and averaged to get the aggregated prediction. Details and complete prompt templates are provided in Appendix A.3. We empirically observe that averaging the verbalized scores over multiple samples for an instance (`Sample-Prob`) helps improve cardinality and performance. Thus, we use a variant of our approach, termed `Ours-Sup-2call`, that takes in two verbalized probabilities (with temperatures 0 and 1) as input. Different from the other supervised approaches, this model is trained on 20 samples per instance and it requires two samples per instance during inference. Note that this is still $10\times$ more efficient compared to `Sample-Prob` during inference.

**Results on the 11 Benchmark Datasets.** Fig. 4 (top) summarizes the experiment results aggregated across 11 datasets. It compares PRAUCs of different approaches calculated over 5 runs with different train/val/test splits on each of the three LLMs, depicting mean and $2-\sigma$ error bars. Detailed results are provided in the Appendix A.9. As previously observed in Section 3.2, proposed methods significantly improve the operational granularity by increasing the diversity and cardinality of the outputs. The proposed methods perform comparably or better than the baseline `Prompt-Naive`. However, a

majority of the 11 datasets are small with as few as 250 samples in five of them. Additionally, the precision-recall curves of the LLMs are mostly monotonic, leaving little scope for improvement. Thus, the performance of proposed supervised methods is comparable to `Ours-Unsup`. Notably, we observe that even though it is significantly simpler, more efficient and scalable than `Sample-Class` and `Sample-Prob`, `Ours-Unsup` performs similarly to the approximated token probabilities on Nova and Qwen LLMs while achieving comparable or higher density in the operational curves.

**Results on a Simulated Diverse Dataset.** Real-world scenarios often involve highly diverse data that contains sub-populations with different distributions. Due to their foundational nature, LLMs are commonly used in these scenarios. To simulate experiments on such a dataset, we combine the 11 datasets we individually experimented with into a single one and analyze the performance. For a single operating point to work well on this combined dataset, it is necessary that the predicted probabilities have consistent meaning across datasets. We observe that the supervised approaches trained on this combined dataset are particularly beneficial in such scenarios. Figure 5 (a): compares the precision-recall curves of different approaches for Claude. For a given approach, we use every unique value in the prediction as a threshold and calculate the corresponding precision and recall and visualize this as a scatter plot. The histograms on the margins denote the number of such points within a bin. We observe that compared to `Prompt-Naive`, proposed method significantly improves the operational granularity by increasing the diversity and cardinality of the outputs, while outperforming it in terms of PRAUC. In addition, we observe that even though it is significantly more efficient and scalable, `Ours-Sup-2call` outperforms `Sample-Prob` while achieving comparable or higher density in the operational curves. Unlike the individual datasets, the performance of `Ours-Unsup` here is not comparable to that of the supervised ones. However, it still offers an easy way to increase the cardinality while outperforming `Prompt-Naive`. Figure 4 (bottom) shows PRAUC comparison for the joint dataset. Unlike on the individual datasets, the proposed supervised approaches and `Sample-Prob` vastly outperform other methods. With just 2 samples per instance during inference, the `Ours-Sup-2call` achieves 8% point improvement in mean PRAUC over the baseline `Prompt-Naive` and 2.6% points over 20 sample `Sample-Prob` approach. The learning approaches have relatively higher variance due to the small sizes of dataset-subsets that are merged to create the combined dataset. Figure 5 (b): shows the operational granularity measurements along $g^{pre}$, $g^{rec}$ and $g^{fpr}$, as well as the output cardinalities. `Prompt-Naive` and the two more complex prompting techniques `Prompt-Seq` and `Prompt-Seq-CoT` have an extremely small cardinality (16 or less). Sampling approaches improve it by nearly two orders of magnitude. However, the proposed methods achieve a cardinality nearly equaling the size of the dataset. This increase in cardinality also translates to better operating points, seen in the low granularity values along all 3 axes of precision, recall and fpr, while maintaining or improving performance.

**Limitations.** All three variants of the proposed method rely on initial LLM predictions with the naive prompting approach. Thus, while this methods can increase the cardinality and provide more choices for operating point selection, it might not always be possible to obtain the optimal interpolation between the naive approaches' points on the operation curves. This is particularly true for the unsupervised approach. The supervised approaches require access to ground-truth labels for training the MLP. While this is akin to typically employed calibration approaches, it adds a layer of complexity over *simple* zero-shot API calls to LLMs. With greater availability of fine-tuning capabilities for black-box LLMs, it might be possible to overcome both these limitations by training them to provide diverse verbalized scores. Additionally, while the proposed methods and analysis can easily be extended to multi-class settings, we limit our experiments to binary classification tasks in this work.

## 5 CONCLUSION

We studied the impact of prompting, confidence elicitation and uncertainty estimation techniques to black-box LLMs' output distributions for decision making problems. We showed that verbalized probabilities have limited cardinality and provided possible explanations for why and when this issue surfaces. Our proposed approaches expand the range of operating points while maintaining or improving the overall prediction quality. Although empirically successful, our approaches do not directly optimize the objective equation 3, but rely on noise to increase granularity. This results in always operating in a granularity-performance tradeoff that may fail to retain performance on tasks with highly complex relationships between $\hat{y}$ and $y$. Future work includes moving away from this tradeoff by modeling $z$ as a prior to a generative model, and fitting a classifier to the representations of this model.

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

# A   Technical Appendices and Supplementary Material

The appendix contains additional discussion on related work in Section A.1, details on datasets in Section A.2 and complete prompts for the approaches we studied in Section A.3. Additional prompting techniques and corresponding results are presented in sections A.4 and A.8. Section A.5 presents an ablation study on prompt specificity. Section A.6 provides additional insights around calibration of LLM-generated probabilities. We provide ablations on model configuration for the proposed approach in Section A.7. Section A.9 contains detailed results for experiments in the main text including dataset-level results.

## A.1   Extended Related Work

**Numerical Capabilities of LLMs.** Our work is centered around the observation that LLMs do not verbalize fine-grained (numerical) probability estimates when prompted. On a higher level, this connects to lines of works that investigate the capabilities of LLMs on domains that involve conducting and expressing probabilistic and mathematical reasoning. Imani et al. (2023) observes that LLMs have limited performance on arithmetic reasoning tasks, and proposes a chain-of-thought (CoT) prompting technique to solve the same math problem in multiple ways to improve the predictions. Paruchuri et al. (2024) investigates the LLMs' capability of estimating percentiles, drawing samples and calculating probabilities. Their experiment results suggest that LLMs have some internal representation that enables probabilistic modeling and reasoning, however, they tend to perform much better on common distributions like uniform and normal compared to others. Although not explicitly discussed, from their experiment results, it can be observed that the verbalized probabilities extracted from LLMs are also low cardinality. Xiong et al. (2023) studied if LLMs can express their own uncertainty and how this expression can be improved by developing prompting, sampling and aggregation strategies. They found that there is not a single strategy that outperforms the others, however, it is recommended to use a combination of top-K prompt, self-random sampling, and average confidence to get stable performance. Notably, these approaches focus on single point performance metrics such as accuracy and expected calibration error (ECE) for evaluation, and they aim to either lift the performance on the task at hand (i.e., fine-tune) or establish consistency between empirical probabilities and model outputs. We instead focus on increasing the granularity of the adjustments we can make to a model's operational behavior. We empirically observe that approaches that involve prompting the LLM multiple times (e.g., CoT and sampling with high temperature) can increase the output diversity, however, this increase is insignificant. See Appendix A.4 for additional experiments with these approaches.

**Model Calibration.** As we study transforming the output distribution (in the form of verbalized probabilities) of black-box LLMs to improve their operational usage, our work can be seen as a type of model calibration. The domain of model calibration studies increasing the reliability and performance of model output distributions, as most machine learning models have inherent biases that impact their predictions. Typically, calibration methods aim to map model predictions to a space where the score distribution has a desired relationship with the empirical ground truth distribution. For example, a probabilistic classifier is considered well-calibrated if the class distribution of the instances that receives the prediction of $p$ is approximately $p$ Silva Filho et al. (2023). Model calibration has been studied extensively from traditional machine learning methods to deep neural networks Minderer et al. (2021); Krishnan & Tickoo (2020). Calibration of LLMs outputs has also been studied in order to mitigate impact of the unwanted bias caused by the choice of prompt Zhou et al. (2023a), architecture LeVine et al. (2023), and pre-training data He et al. (2024). Notably, most of the existing works for calibrating LLMs rely on access to token probabilities Xie et al. (2024). In black-box LLMs, it is more common to conduct post-processing Detommaso et al. (2024) or training auxiliary models using the same input as the LLM Ulmer et al. (2024) (i.e., converting the setting to gray-box). Existing post-processing methods focus on regulating the coarse-grained verbalized confidence scores and do increase the cardinality of these scores. Hence, they are not immediately applicable to enriching performance curves to set operating points on. On the other hand, the auxiliary models rely on training on the same input provided to the LLM for inference. This improves the operational granularity of the whole system, given that the system now includes a score generating models that is trained on the given task in a fully-supervised setting (as it involves training another language model to be used with the LLM). However, auxiliary approaches significantly increase the amount of annotated data and the level of expertise needed to reliably generalize, compared to using black-box LLMs as they

are. Additional, these approaches suffer from the increased length and complexity of the inputs, as their predictions are conditioned on the input of the black box LLMs. In this work, we explicitly limit our scope to simple, efficient and scalable approaches and avoid using the signals from the input features directly. This helps us to focus on correcting LLM behavior that generalizes across samples and isolate the complexity of extracting features from textual input within the LLM, instead of learning corrections conditioned over input features.

## A.2 EXPERIMENT DETAILS

**Datasets.** Table 2 lists the dataset names, with their numbers of samples and URLs. We drop `SST2` and `BoolQ`'s test sets as their labels are not publicly available. We also drop the `passage` field of the `Boolq` dataset as the context it provides makes the task significantly easier for the LLMs we experiment with. `{Heart, Income, Jungle}-Serialized-TabLLM` are tabular datasets that were serialized to natural-language strings using by Hegselmann et al. (2023). We include these datasets to introduce variety to the set of tasks we experiment with, as they frequently involve decision making using numerical features together with categorical ones. When the dataset has its own test split available, we directly use it for testing. Otherwise, we combine all the available splits and randomly sample 20% of the combined splits to create a set split. With this, we end up with train and test splits with 145262 and 42084 samples, respectively. With single-LLM-call approaches, we first experiment with these splits and observe that low cardinality and rounding bias are prominent problems across $> 180k$ samples, and magnitudes of these issues are only sub-linearly related to the number of samples (Figure 6). With this observation, as we expand our baselines to include additional single and multi-call approaches that require up to 20 LLM calls per sample, we reduce redundancy of LLM calls and use the original test split (42084 samples) as our full dataset, and further split it into new training and test splits (21042 samples each). For each learning-based approach, we randomly sample this new training set by %20 for validation. All experiments presented in the main body of this paper use these new training and test splits. In order to simulate a diverse dataset, we combine the 11 datasets by respecting these training and test splits. This simulated task represents a heterogeneous classification use case where sub-populations have different distributions.

**Hardware and Services.** We use Amazon Bedrock's Amazon (2025a) APIs to experiment with `Anthropic Claude 3.0 Sonnet` and `Amazon Nova Pro 1.0`. We experiment with `DeepSeek R1 Distill-Qwen-32B` by deploying the model to 9 Amazon SageMaker Amazon (2025c) endpoints with `g5.48xlarge` compute instances and running inference on the in parallel. We conduct all non-LLM-inference compute (including the experiments with proposed approaches) using two `c5.12xlarge` Amazon SageMaker notebook instances Amazon (2025c).

**Implementation of proposed-unsupervised**. We provide the implementation of the unsupervised variant of the proposed approach below. The implementation assumes that the LLM verbalized probabilities are provided in a DataFrame `df`. Then, for each row, it samples from a uniform distribution with its range determined by the corresponding row's verbalized probability (i.e., `1-score`) and the next larger verbalized prediction that was previously generated.

```
# proposed-unsupervised start

import pandas as pd
import numpy as np

def find_smallest_larger(x, l):
    '''Find the smallest score in the list of past predictions l,
    that is larger than x.'''

    smallest_larger = None

    for num in l:
        if num > x:
            if smallest_larger is None or num < smallest_larger:
                smallest_larger = num

    return smallest_larger

def proposed_noise(pred, all_scores):
    '''Sample the additive noise from a range that
    ensures that when added to pred, it won't change
    the order of predictions.'''

    upper_bound = find_smallest_larger(pred, all_scores)
    if upper_bound == None:
```

```
      noise_upper_bound = 0

   else:
      noise_upper_bound = np.max((0, (upper_bound - pred) - 1e-9))

   noise = np.random.uniform(0, noise_upper_bound)
   return noise

"""Let df be a Pandas DataFrame with LLM verbalized
predictions provided in the column '1-score'"""

all_scores = list(set(df['1-score'].unique()) | {0, 1})
df['noise_proposed'] = df['1-score'].apply(lambda x: proposed_noise(x, all_scores))
noisy_predictions = df['1-score'] + df['noise_proposed']

# proposed-unsupervised end
```

**Implementation of proposed-supervised.** We use `PyTorch` and `PyTorch Lightning` to implement the supervised variant of the proposed method as a simple MLP. We avoid searching for width and depth of the MLP, and simply set the implementation to always use a 2-hidden layer MLP with layer widths set to $2^{insize}$. For easy extension to multi-class, we prompt the LLM to output verbalized probabilities for each class. Hence, for binary classification, this translates to an $insize$ of 3 (2 from verbalized probabilities, 1 from sampled noise). For the 2-call variant of the proposed method, the $insize$ gets increased by 2 as there are 2 more verbalized probabilities added from the additional `temperature=1` call. Finally, we limit the maximum number of training epochs to 50, and use early stopping on the validation split PRAUC with patience set to 5. We truncate the logging steps from the code provided below for readability.

```
# proposed-supervised start

import torch
import lightning as L
from torch import optim, nn

class Proposed(L.LightningModule):
   def __init__(self, insize, outsize, learning_rate, bias_weight=1.0, lambda_bias=0.01):
      super().__init__()

      self.mlp = nn.Sequential(
               nn.Linear(insize-1, 2**insize),
               nn.ReLU(),
               nn.Linear(2**insize, 2**insize),
               nn.ReLU(),
               nn.Linear(2**insize, outsize),
            )

      self.bias_weight = nn.Parameter(torch.tensor(bias_weight))
      self.learning_rate = learning_rate
      self.lambda_bias = lambda_bias

   def training_step(self, batch, batch_idx):
      x, y = batch
      x, bias = x[:, :-1], x[:, -1:]
      yhat = nn.functional.sigmoid(self.mlp(x) + bias * 1/self.bias_weight)
      loss = nn.functional.binary_cross_entropy(yhat, y.unsqueeze(1)) + torch.abs(self.bias_weight) *
            self.lambda_bias
      return loss

   def validation_step(self, batch, batch_idx):
      x, y = batch
      x, bias = x[:, :-1], x[:, -1:]
      yhat = nn.functional.sigmoid(self.mlp(x) + bias * 1/self.bias_weight)
      loss = nn.functional.binary_cross_entropy(yhat, y.unsqueeze(1)) + torch.abs(self.bias_weight) *
            self.lambda_bias
      return loss

   def forward(self, x):
      x, bias = x[:, :-1], x[:, -1:]
      return nn.functional.sigmoid(self.mlp(x) + bias * 1/self.bias_weight)

   def configure_optimizers(self):
      optimizer = optim.Adam(self.parameters(), lr=self.learning_rate)
      return optimizer

# proposed-supervised start
```

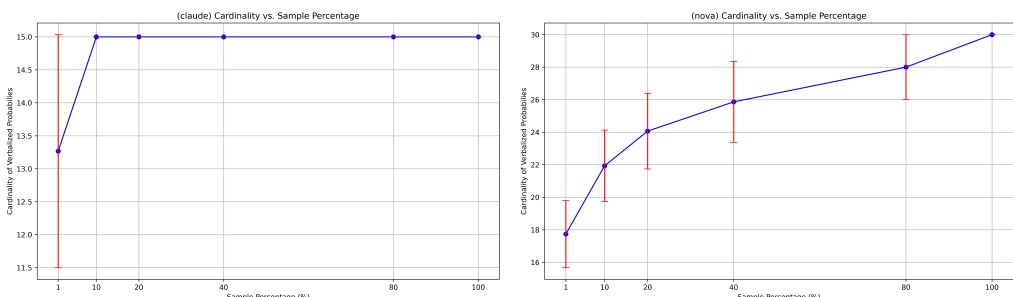

Figure 6: Cardinality vs. sampling percentage for Claude (left) and Nova (right). We observe that there is a sub-linear relationship between the number of samples and the cardinality they correspond to. Error bar represent $2 - \sigma$ ranges calculated across 15 random seeds.

| Dataset | #Samples | Source URL |
|---|---|---|
| SST2 (train., val.) | 68221 | huggingface.co/datasets/stanfordnlp/sst2 |
| Movie Reviews | 10662 | huggingface.co/datasets/cornell-movie-review-data/rotten_tomatoes |
| Heart-Serialized-TabLLM Hegselmann et al. (2023) | 918 | github.com/clinicalml/TabLLM/tree/main/datasets_serialized/heart |
| Income-Serialized-TabLLM Hegselmann et al. (2023) | 48842 | github.com/clinicalml/TabLLM/tree/main/datasets_serialized/income |
| Jungle-Serialized-TabLLM Hegselmann et al. (2023) | 44819 | github.com/clinicalml/TabLLM/tree/main/datasets_serialized/jungle |
| BoolQ (train., dev.) | 12697 | https://github.com/google-research-datasets/boolean-questions |
| BIG-Bench Hard-Boolean Expressions | 250 | github.com/suzgunmirac/BIG-Bench-Hard/tree/main/bbh/boolean_expressions.json |
| BIG-Bench Hard-Causal Judgement | 187 | github.com/suzgunmirac/BIG-Bench-Hard/tree/main/bbh/causal_judgement.json |
| BIG-Bench Hard-Formal Fallacies | 250 | github.com/suzgunmirac/BIG-Bench-Hard/tree/main/bbh/formal_fallacies.json |
| BIG-Bench Hard-Sports Understanding | 250 | github.com/suzgunmirac/BIG-Bench-Hard/tree/main/bbh/sports_understanding.json |
| BIG-Bench Hard-Hyperbaton | 250 | github.com/suzgunmirac/BIG-Bench-Hard/tree/main/bbh/hyperbaton.json |

Table 2: Dataset sizes and sources used in experiments.

## A.3 LIST OF PROMPTS USED IN EXPERIMENTS

### A.3.1 BASELINE PROMPT TEMPLATE

We use the below prompt template for all our experiments with both the baseline and proposed approaches.

```
Prompt:
"""
<task>
    {context}
</task>

<input_sentence>
    {input}
</input_sentence>

<formatting instructions>
    - Provide your final answer **only** in the specified JSON format below.
    - Do **not** include any explanations or additional text outside the JSON.
    - Ensure the JSON is valid and properly formatted.
    - Do **not** include any extra characters or text before or after the JSON.
</formatting instructions>

Your output should look like this but in JSON format:
<expected-output>
    {category_probabilities}
    <reason>Explain your reasoning.</reason>
    <decision>Return the most probable category for the input.</decision>
    <decision-confidence>Provide the probability of your decision being correct
    in the range of 0 to 1.</decision-confidence>
</expected-output>
"""
```

### A.3.2 LOW SPECIFICITY PROMPT TEMPLATE

```
Specificity: Low
Prompt:
"""
<task>
    {context}
</task>

<input_sentence>
    {input}
</input_sentence>
```

```
<formatting instructions>
   - Provide your final answer **only** in the specified JSON format below.
   - Do **not** include any explanations or additional text outside the JSON.
   - Ensure the JSON is valid and properly formatted.
   - Do **not** include any extra characters or text before or after the JSON.
</formatting instructions>

While generating your output, follow the instructions provided below:
<task_instructions>
1. Your task is to estimate the class probabilities given input. In order to do this, determine the input
     features that will contribute to your decision.
2. Then, based on your understanding of the task and the input features you selected, pick a function
     (hypothesis) that takes these input features in, and outputs a continuous decision estimate.
3. Using your input features, hypothesis and its weights, calculate the classification score as your output.
</task_instructions>

Your output should look like this but in JSON format:
<expected-output>
   <selected-features>List the names and values of the features you selected to generate your output
       with.</selected-features>
   <hypothesis-function>Describe the function you picked for decision making.</hypothesis-function>
   {category_probabilities}
   <reason>Explain your reasoning.</reason>
   <decision>Return the most probable category for the input.</decision>
   <decision-confidence>Provide the probability of your decision being correct in the range of 0 to
       1.</decision-confidence>
</expected-output>
"""
```

### A.3.3  MEDIUM SPECIFICITY PROMPT TEMPLATE

```
Specificity: Medium
Prompt:
"""
<task>
   {context}
</task>

<input_sentence>
   {input}
</input_sentence>

<formatting instructions>
   - Provide your final answer **only** in the specified JSON format below.
   - Do **not** include any explanations or additional text outside the JSON.
   - Ensure the JSON is valid and properly formatted.
   - Do **not** include any extra characters or text before or after the JSON.
</formatting instructions>

While generating your output, follow the instructions provided below:
<task_instructions>
1. Your task is to estimate the class probabilities given input. In order to do this, determine the input
     features that will contribute to your decision and extract their values. You have to ALWAYS assign a
     continuous numerical value from [0-1] with high precision (can have many decimal points) to each
     feature you picked, and work with these values in the next steps to represent the features.
2. Then, based on your understanding of the task and the input features you selected, pick a function
     (hypothesis) that takes these input features in, and outputs a continuous decision estimate.
3. Using your input features and the hypothesis function and calculate the classification score as your
     output.
</task_instructions>

Your output should look like this but in JSON format:
<expected-output>
   <selected-features>List the names and values of the features you selected to generate your output
       with.</selected-features>
   <hypothesis-function>Describe the function you picked.</hypothesis-function>
   {category_probabilities}
   <reason>Explain your reasoning.</reason>
   <decision>Return the most probable category for the input.</decision>
   <decision-confidence>Provide the probability of your decision being correct in the range of 0 to
       1.</decision-confidence>
</expected-output>
"""
```

### A.3.4  HIGH SPECIFICITY PROMPT TEMPLATE

```
Specificity: High
Prompt:
"""
<task>
   {context}
</task>

<input_sentence>
   {input}
```

```
</input_sentence>

<formatting instructions>
    - Provide your final answer **only** in the specified JSON format below.
    - Do **not** include any explanations or additional text outside the JSON.
    - Ensure the JSON is valid and properly formatted.
    - Do **not** include any extra characters or text before or after the JSON.
</formatting instructions>

While generating your output, follow the instructions provided below:
<task_instructions>
1. Your task is to estimate the class probabilities given input. In order to do this, determine the input
    features that will contribute to your decision and extract their values. You have to ALWAYS assign a
    continuous numerical value from [0-1] with high precision (can have many decimal points) to each
    feature you picked, and work with these values in the next steps to represent the features.
2. Then, based on your understanding of the task and the input features you selected, pick a function
    (hypothesis) parameterized by weights for each input feature, and that takes these input features in,
    and outputs a continuous decision estimate.
3. Assign continuous weights to your hypothesis function considering the desired impact of each input feature
    to the decision.
4. Next, having chosen the input features and weights of your prediction fuction, calculate your continuous
    decision estimates (i.e., probability of classes) as your output by running input features through the
    function you picked.
</task_instructions>

Your output should look like this but in JSON format:
<expected-output>
    <selected-features>List the names and values of the features you selected to generate your output
        with.</selected-features>
    <hypothesis-function>Describe the function you picked.</hypothesis-function>
    <weights>List the weights you assigned to each input feature to parameterize the logistic regression
        function.</weights>
    <calculation>Break down the calculation of your outputs using selected feature values, hypothesis
        functions and weights.</calculation>
    {category_probabilities}
    <reason>Explain your reasoning.</reason>
    <decision>Return the most probable category for the input.</decision>
    <decision-confidence>Provide the probability of your decision being correct in the range of 0 to
        1.</decision-confidence>
</expected-output>
"""
```

## A.3.5  LINEAR AND LOGISTIC PROMPT TEMPLATES USED IN APPENDIX A.5

```
Specificity: Linear
Prompt:
"""
<task>
    {context}
</task>

<input_sentence>
    {input}
</input_sentence>

<formatting instructions>
    - Provide your final answer **only** in the specified JSON format below.
    - Do **not** include any explanations or additional text outside the JSON.
    - Ensure the JSON is valid and properly formatted.
    - Do **not** include any extra characters or text before or after the JSON.
</formatting instructions>

While generating your output, follow the instructions provided below:
<task_instructions>
1. Your task is to estimate the class probabilities given input. In order to do this, determine the input
    features that will contribute to your decision and extract their values. You have to ALWAYS assign a
    continuous numerical value from [0-1] with high precision (can have many decimal points) to each
    feature you picked, and work with these values in the next steps to represent the features.
2. Then, based on your understanding of the task and the input features you selected, use a linear weighted
    combination that combines the input features (hypothesis function), and outputs a continuous decision
    estimate.
3. Assign continuous weights to your hypothesis function considering the desired impact of each input feature
    to the decision.
4. Next, having chosen the input features and weights of your prediction fuction, calculate your continuous
    decision estimates (i.e., probability of classes) as your output by running input features through the
    function you picked. If your output is not in [0, 1], you are allowed to round to the closest value
    inside this range.
</task_instructions>

Your output should look like this but in JSON format:
<expected-output>
    <selected-features>List the names and values of the features you selected to generate your output
        with.</selected-features>
    <hypothesis-function>Describe the function you picked.</hypothesis-function>
    <weights>List the weights you assigned to each input feature to parameterize the logistic regression
        function.</weights>
    <calculation>Break down the calculation of your outputs using selected feature values, hypothesis
        functions and weights.</calculation>
    {category_probabilities}
```

```
    <reason>Explain your reasoning.</reason>
    <decision>Return the most probable category for the input.</decision>
    <decision-confidence>Provide the probability of your decision being correct in the range of 0 to
        1.</decision-confidence>
</expected-output>
"""
```

```
Specificity: Logistic
Prompt:
"""
<task>
    {context}
</task>

<input_sentence>
    {input}
</input_sentence>

<formatting instructions>
    - Provide your final answer **only** in the specified JSON format below.
    - Do **not** include any explanations or additional text outside the JSON.
    - Ensure the JSON is valid and properly formatted.
    - Do **not** include any extra characters or text before or after the JSON.
</formatting instructions>

While generating your output, follow the instructions provided below:
<task_instructions>
1. Your task is to estimate the class probabilities given input. In order to do this, determine the input
    features that will contribute to your decision and extract their values. You have to ALWAYS assign a
    continuous numerical value from [0-1] with high precision (can have many decimal points) to each
    feature you picked, and work with these values in the next steps to represent the features.
2. Then, based on your understanding of the task and the input features you selected, use a linear weighted
    combination that combines the input features (hypothesis function) followed by a logistic sigmoid
    function (1/(1+e^(-x))) to output a continuous decision estimate.
3. Assign continuous weights to your hypothesis function considering the desired impact of each input feature
    to the decision.
4. Next, having chosen the input features and weights of your prediction fuction, calculate your continuous
    decision estimates (i.e., probability of classes) as your output by running input features through the
    function you picked.
</task_instructions>

Your output should look like this but in JSON format:
<expected-output>
    <selected-features>List the names and values of the features you selected to generate your output
        with.</selected-features>
    <hypothesis-function>Describe the function you picked.</hypothesis-function>
    <weights>List the weights you assigned to each input feature to parameterize the logistic regression
        function.</weights>
    <calculation>Break down the calculation of your outputs using selected feature values, hypothesis
        functions and weights.</calculation>
    {category_probabilities}
    <reason>Explain your reasoning.</reason>
    <decision>Return the most probable category for the input.</decision>
    <decision-confidence>Provide the probability of your decision being correct in the range of 0 to
        1.</decision-confidence>
</expected-output>
"""
```

### A.3.6 PROMPT CONTEXTS

Dataset specific information (e.g., task description) is added to the prompt using the `context` and `category_probabilities` arguments. The values for these for each dataset are provided below. It can be seen that we keep the prompt contexts minimal, just to ensure that the model is set to conduct the evaluation correctly. We do not use these contexts to provide additional information to the model that can explicitly impact their output distributions.

```
'sst2': {
'context': """
You are given sentences from movie reviewes.
Your task is to determine if the sentiment of a given sentence is positive or
negative.
If the sentiment is positive, return 'positive'.
If the sentiment is negative, return 'negative'.
These are the only acceptable answers.""",
},
'mr': {
'context': """
You are given sentences from Rotten Tomatoes movie reviewes.
Your task is to determine if the sentiment of a given sentence is positive or
negative.
If the sentiment is positive, return 'pos'.
If the sentiment is negative, return 'neg'.
These are the only acceptable answers.""",
},
'heart': {
```

```
'context': """
Given an input, answer the following question: does the coronary angiography of
this patient show a heart disease?
'yes' and 'no' are the only acceptable answers.""",
},
'income': {
'context': """
Given an input, answer the following question: does this person earn more than
50000 dollars per year?
'yes' and 'no' are the only acceptable answers.""",
},
'jungle': {
'context': """
Given an input, answer the following question: does the white player win this
two pieces endgame of Jungle Chess?
'yes' and 'no' are the only acceptable answers.""",
},
'boolq': {
'context': """Answer the question given below. 'yes' and 'no' are the only
acceptable answers.""",
},
'bigbenchhard_boolean_expressions': {
'context': """Solve the following boolean expression. 'True' and 'False' are the
only acceptable answers.""",
},
'bigbenchhard_causal_judgement': {
'context': """Answer the following question.""",
},
'bigbenchhard_formal_fallacies': {
'context': """Answer the following question.""",
},
'bigbenchhard_sports_understanding': {
'context': """Answer the question given below. 'yes' and 'no' are the only
acceptable answers.""",
},
'bigbenchhard_hyperbaton': {
'context': """Answer the question given below. '(A)' and '(B)' are the only
acceptable answers.""",
}
```

We set the field {category_probabilities} using the acceptable answers provided above,
and the function below.

```
def set_category_probabilities(dataset_name):
    prompt = """"""
    values = list(metadata[dataset_name]['labelmap'].values()) # labelmap contains the list of all possible
        labels of the corresponding dataset
    for value in values:
        prompt += f"<{value}-score> Return the probability of input belonging to the category '{value}', from 0
            to 1, 1 corresponding to the strongest chance of belonging. </{value}-score>\n"

    return prompt
```

## A.4 Additional Experiments with Simple Prompting Techniques to Extract Verbalized Probabilities

As discussed in Section 3.2 of the main text, we consider several prompting techniques to help improve the LLM output cardinality. We describe those in detail here along with their complete prompts and experimental results on `Anthropic Claude 3.0 Sonnet`. All the methods below use a single call to the LLM per input instance.

`Score-0-100`: In the baseline prompting approach `Prompt-Naive`, we predict the output in the range $[0, 1]$. On the blackbox Qwen LLM, we observe that the LLM always predicts the output digit by digit by using a separate token for each of them. We thus also experiment with an output range of $[0, 100]$.

`Non-Mulitple-of-5`: Section 3.2 of main text shows that LLMs tend to predict probability values in steps of $0.05$, resulting in a decrease in output cardinality. Here, the model is encouraged to predict non-multiples of $0.05$ as the output.

`Two-Decimal-Digits`: We observe that a majority of the LLM predictions with `Prompt-Naive` have just a single decimal digit. With this prompt, the model is explicitly asked to predict an output with two decimal digits.

`Coarse-Fine`: The model is asked to arrive at the final probability as the sum of a coarse initial prediction and an offset in the range $[-0.03, 0.03]$. If the coarse predictions are limited to multiples of $0.05$, we expect the final outputs to be more diverse due to diversity in the offset values.

| Approach | $\|\hat{y}\| \uparrow$ | $g^{pre} \downarrow$ | $g^{rec} \downarrow$ | $g^{fpr} \downarrow$ | PRAUC $\uparrow$ | AUROC $\uparrow$ |
|---|---|---|---|---|---|---|
| `Prompt-Naive` | 10 | 0.145 | 0.426 | 0.373 | 0.75 | 0.80 |
| `Score-0-100` | 10 | 0.200 | 0.514 | 0.333 | 0.74 | 0.79 |
| `Non-Mulitple-of-5` | 13 | 0.150 | 0.423 | 0.355 | 0.75 | 0.79 |
| `Two-Decimal-Digits` | 17 | 0.150 | 0.335 | 0.269 | 0.75 | 0.79 |
| `Coarse-Fine` | 21 | 0.139 | 0.412 | 0.295 | 0.75 | 0.79 |
| `In-Context` | 13 | 0.172 | 0.402 | 0.357 | 0.75 | 0.80 |
| `Multiple-Pred` | 70 | 0.098 | 0.100 | 0.122 | 0.73 | 0.76 |
| lightpurple `Ours-Unsup` | 5614 | 0.079 | 0.058 | 0.055 | 0.76 | 0.80 |

Table 3: Aggregated results on 11 individual datasets. Operational granularity of different prompting techniques measured along precision ($pre$), recall ($rec$) and false positive rate ($fpr$) axes to cover both PR and ROC spaces, as well as their output cardinalities ($car$). Granularities are calculated using Equation equation 2 of main text. All methods except for `Ours-Unsup` rely only on prompting to try to improve the output cardinality while `Ours-Unsup` uses additive Gaussian noise atop `Prompt-Naive`. None of the prompting techniques significantly improve either the cardinality or, more importantly, granularity compared to the naive prompting baseline. Generating multiple predictions and randomly choosing one of them (`Multiple-Pred`) offers the best improvement in terms of cardinality.

`In-Context`: LLMs are known to have great in-context learning ability. We provide a diverse set of probability values as context and instruct the LLM to generate similar outputs.

`Multiple-Pred`: Experiments in the main paper (Section 4) suggest that sampling an LLM multiple times for the same input instance results in better distribution of output probability values. We consider a more efficient version of it here where the model predicts multiple values for a single input prompt. The values are then aggregated to obtain the final prediction. We observe that just randomly choosing one of the predicted values helps improve cardinality the most compared to other methods like mean and median.

Tables 3 and 4 compare the different prompting techniques along with the proposed `Ours-Unsup` method that uses additive noise on both the 11 individual datasets and the joint dataset. We report the averaged metrics for the 11 datasets results. While the tailored prompts do increase the cardinality compared to `Prompt-Naive`, all of them still have very low cardinalities compared to `Ours-Unsup`. The increase in cardinality also does not necessarily reflect in improved granularities, limiting their utility. Predicting multiple outputs and choosing one of them randomly (`Multiple-Pred`) results in the best performance in terms of operational granularity but results in degradation of PRAUC and ROC-AUC, particularly on the individual datasets. Note that due to the extremely low cardinality of these methods, the PRAUC and ROC-AUC tend to be over-estimates due to linear interpolation between points.

### A.4.1 PROMPT TEMPLATES

The prompts for each of the methods are provided below. Since there are no changes in the input and context parts of the prompt compared to the baseline prompt (A.3.1), we only provide the part of the prompt corresponding to the output here.

Score Range [0, 100] Prompt Template

```
Prompt:
"""
    Your output should look like this but in JSON format:
    <expected-output>
        {category_probabilities}
        <reason>Explain your reasoning.</reason>
        <decision>Return the most probable category for the input.</decision>
        <decision-confidence>Provide the probability of your decision being correct in the range of 0 to
            1.</decision-confidence>
    </expected-output>
"""

def set_category_probabilities(dataset_name):
    prompt = """"""
```

| Approach | $\|\hat{y}\|$ ↑ | $g^{pre}$ ↓ | $g^{rec}$ ↓ | $g^{fpr}$ ↓ | PRAUC ↑ | AUROC ↑ |
|---|---|---|---|---|---|---|
| `Prompt-Naive` | 15 | 0.081 | 0.444 | 0.198 | 0.72 | 0.77 |
| `Score-0-100` | 15 | 0.062 | 0.409 | 0.170 | 0.70 | 0.76 |
| `Non-Mulitple-of-5` | 34 | 0.114 | 0.344 | 0.258 | 0.74 | 0.79 |
| `Two-Decimal-Digits` | 41 | 0.068 | 0.216 | 0.150 | 0.72 | 0.74 |
| `Coarse-Fine` | 45 | 0.086 | 0.194 | 0.168 | 0.74 | 0.80 |
| `In-Context` | 24 | 0.076 | 0.413 | 0.240 | 0.72 | 0.79 |
| `Multiple-Pred` | 103 | 0.033 | 0.075 | 0.081 | 0.74 | 0.75 |
| lightpurple `Ours-Unsup` | 39294 | 0.020 | 0.035 | 0.010 | 0.72 | 0.77 |

Table 4: Results on joint dataset. Operational granularity of different prompting techniques measured along precision ($pre$), recall ($rec$) and false positive rate ($fpr$) axes to cover both PR and ROC spaces, as well as their output cardinalities ($car$). Granularities are calculated using Equation equation 2 of main text. All methods except for `Ours-Unsup` rely only on prompting to try to improve the output cardinality while `Ours-Unsup` uses additive Gaussian noise atop `Prompt-Naive`. Similar to individual datasets scenario, none of the prompting techniques significantly improve cardinality or operational granularity. `Multiple-Pred` again has the highest cardinality amongst the prompting techniques while predicting the score first as a coarse value and then predicting an offset to get the final value (`Coarse-Fine`) achieves the best performance in terms of PRAUC and ROC-AUC. However, note that due to the extremely low cardinality of these methods, the PRAUC and ROC-AUC tend to be over-estimates due to linear interpolation between points.

```python
values = list(metadata[dataset_name]['labelmap'].values())
for value in values:
    prompt += f"<{value}-score> Return the likelihood of input belonging to the category '{value}', from 0
        to 100, 100 corresponding to the strongest chance of belonging. </{value}-score>\n"
return prompt
```

### Non-multiples of 5 Prompt Template

```
Prompt:
"""
    Your output should look like this but in JSON format:
    <expected-output>
        {category_probabilities}
        For the probabilities, do not just default to multiples of 0.05. Over a dataset, the values must have
            enough spread while being comparable across samples to provide an operating point for any desired
            precision or recall.
        <reason>Explain your reasoning.</reason>
        <decision>Return the most probable category for the input.</decision>
        <decision-confidence>Provide the probability of your decision being correct in the range of 0 to
            1.</decision-confidence>
    </expected-output>
"""
```

### Two Decimal Digits Prompt Template

```
Prompt:
"""
    Your output should look like this but in JSON format:
    <expected-output>
        {category_probabilities}
        Predict the probabilities with two decimal places.
        <reason>Explain your reasoning.</reason>
        <decision>Return the most probable category for the input.</decision>
        <decision-confidence>Provide the probability of your decision being correct in the range of 0 to
            1.</decision-confidence>
    </expected-output>
"""
```

### Coarse-to-fine Prompt Template

```
Prompt:
"""
    Your output should look like this but in JSON format:
    <expected-output>
        {category_probabilities}
        Obtain the probability values as a coarse-grained and a fine-grained value. The fine-grained value must
            be within 0.03 of the coarse-grained prediction. Over a dataset, the fine-grained values must
```

```
            have enough spread while being comparable across samples to provide an operating point for any
            desired precision or recall.
        <reason>Explain your reasoning.</reason>
        <decision>Return the most probable category for the input.</decision>
        <decision-confidence>Provide the probability of your decision being correct in the range of 0 to
            1.</decision-confidence>
    </expected-output>
"""

def set_category_probabilities(dataset_name):
    prompt = """"""
    values = list(metadata[dataset_name]['labelmap'].values())
    for value in values:
        prompt += f"<{value}-score-coarse>\n <{value}-score> Return the probability of input belonging to the
            category '{value}', from 0 to 1, 1 corresponding to the strongest chance of belonging.
            </{value}-score>\n"
    return prompt
```

### In-context Prediction Prompt Template

```
Prompt:
"""
    Your output should look like this but in JSON format:
    <expected-output>
        {category_probabilities}
        Over a dataset, the probability values must have enough spread while being comparable across samples to
            provide an operating point for any desired precision or recall. Here is an example of predicted
            probabilities for a class over 25 samples in sorted order: [0.01, 0.03, 0.05, 0.08, 0.1, 0.12,
            0.19, 0.25, 0.28, 0.32, 0.4, 0.46, 0.53, 0.6, 0.66, 0.71, 0.75, 0.77, 0.81, 0.84, 0.88, 0.9,
            0.92, 0.95, 0.96, 0.99].
        <reason>Explain your reasoning.</reason>
        <decision>Return the most probable category for the input.</decision>
        <decision-confidence>Provide the probability of your decision being correct in the range of 0 to
            1.</decision-confidence>
    </expected-output>
"""
```

### Multiple Predictions Prompt Template

```
Prompt:
"""
    Your output should look like this but in JSON format:
    <expected-output>
        {category_probabilities}
        Current probability value prediction must not be dependent on the previous predicted values.
        <reason>Explain your reasoning.</reason>
        <decision>Return the most probable category for the input.</decision>
        <decision-confidence>Provide the probability of your decision being correct in the range of 0 to
            1.</decision-confidence>
    </expected-output>
"""

def set_category_probabilities(dataset_name):
    prompt = """"""
    values = list(metadata[dataset_name]['labelmap'].values())
    for value in values:
        prompt += f"<{value}-score> Return a list of 20 independent predictions of probability of input
            belonging to the category '{value}', from 0 to 1, 1 corresponding to the strongest chance of
            belonging.</{value}-score>\n"
    return prompt
```

### A.5 DISCUSSION ON PROMPT SPECIFICITY AND ABLATIONS

We first provide an informal discussion on how we think about prompt specificity. We look at LLM inference from a lens where the LLM is assumed to be a sophisticated search engine that retrieves a *good* combination of tokens given an input prompt. Then, a prompt can be seen as a set of requirements (query) to reduce the space of all answers to the space of acceptable answers. Different from traditional search engines, LLMs conduct this operation without constructing all answers completely, but by sequentially choosing tokens in a greedy manner according to the token probabilities internally encoded. Then, if the prompt is not specific enough (i.e., there are many tokens that fit the constraints provided in the prompt), the LLM's choice among the acceptable candidates can be influenced by its inherent biases (such as the rounding bias). Increasing the specificity of the prompt may help mitigating these biases, however, it is not trivial to craft a prompt that only targets these biases.

In this section, we argue that correctly introducing specificity to only increase the cardinality of verbalized probabilities is not a straightforward task. With these two attempts, we aim to understand if

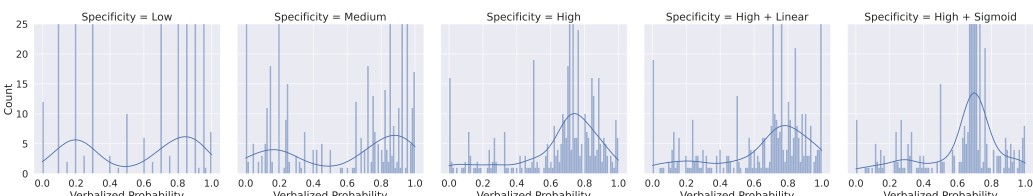

Figure 7: Verbalized probability distributions of Qwen using prompts with low, medium and high specificity, together with . The output cardinality can be increased by providing additional instructions on how to conduct the task.

| Prompt Specificity | $|\hat{y}|$ ↑ | $g^{\text{pre}}$ ↓ | $g^{\text{rec}}$ ↓ | $g^{\text{fpr}}$ ↓ | PRAUC↑ | AUROC↑ |
|---|---|---|---|---|---|---|
| Low | 28 | 0.091 | 0.188 | 0.297 | 0.775 | 0.792 |
| Medium | 89 | 0.031 | 0.117 | 0.104 | 0.713 | 0.748 |
| High | 179 | 0.010 | 0.050 | 0.042 | 0.633 | 0.663 |
| Linear | 120 | 0.014 | 0.081 | 0.067 | 0.647 | 0.678 |
| Logistic | 128 | 0.033 | 0.063 | 0.067 | 0.666 | 0.689 |

Table 5: Cardinality, granularity and AUCs of Qwen's verbalized probability distributions when prompted with Low, Medium, High specificity, together with the attempts *Linear* and *Logistic*.

the nature of the additional instructions introduced plays a role in the change in cardinality. We extend the prompt specificity experiment results depicted in Figure 3 by introducing two more attempts that provide additional details on how to conduct the task: *Linear* and *Logistic* (A.3.5). *Linear* contains the same instructions as the *High Specificity* prompt A.3.4, however, it explicitly prompts the LLM to use a weighted linear combination of features, and truncate the results to [0, 1] range. Similarly, *Logistic* extends *Linear* by instructing the LLM to use a logistic sigmoid function after the weighted linear combination. Specifically, both *Linear* and *Logistic* attempt to increase specificity by instructing the LLM to use specific function classes, potentially limiting LLMs flexibility to pick different functions per input. This is different than the specificity introduced in *Medium* and *High* where the prompt instructs the LLM to explicitly work with continuous numerical features and weights to generate a prediction, which involves having to conduct explicit mathematical operations that likely result in non-round numbers (even if the weights and features are round).

Figure 7 and Table 5 depict the extended experiment results. We observe that the two new attempts `Linear` and `Logistic` result in reduced cardinality and degraded operational granularity compared to the high specificity prompt. This is because, even though these two attempts provide additional instructions, they also impact the LLM's behavior even when it is not biased towards producing low cardinality outputs (i.e., they change and restrict the function families that LLM can use).

We conclude that there is a complex relationship between the prompt used and the cardinality of the verbalized probabilities it produces, and it is not trivial to design prompts that correctly reduce ambiguity (i.e., increase specificity), as the notion of ambiguity is a function of the LLM's knowledge and biases, as well as the dataset. This makes prompt tuning an unreliable approach to deal with limited verbalized probability cardinality and diversity.

## A.6 CALIBRATION OF LLM-GENERATED PROBABILITY ESTIMATES

We extend our discussion from Section 3.2 of the main paper and investigate if using verbalized probabilities instead of the internal token probabilities change the calibration behavior. Notably, this subject has conflicting findings in the existing literature. For example, Lin et al. (2022) found that both logits and verbalized probabilities are well-calibrated. Ulmer et al. (2024) found that using verbalized probabilities is worse than using the likelihood of the corresponding token sequence in most of the settings. On the other hand, Tian et al. (2023) found that verbalized confidences are

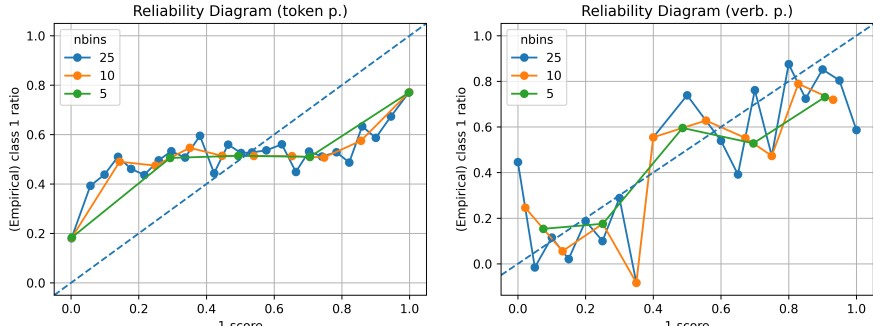

Figure 8: Reliability diagrams of Qwen across 11 datasets using token probabilities (token. p.) and verbalized probabilities (verb. p.)

typically better calibrated than the model's conditional probabilities. We use the same 11 datasets from Section 3.2 (main paper), and compare the reliability diagrams Guo et al. (2017) of `DeepSeek R1 Distill-Qwen-32B`'s token and verbalized probabilities in Figure 8. In our setting, we find that using token probabilities provide a smoother and more monotonously increasing calibration curve compared to using the verbalized probabilities due to their high cardinality. On the other hand, verbalized probabilities have lower expected calibration error (ECE) in the low and high score regions. We conclude that verbalization of probabilities does not significantly harm ECE, and may help mitigating under/over confidence of the LLMs in low/high score regions.

## A.7 ABLATIONS ON THE PROPOSED METHOD DESIGN

In the proposed supervised approach `Ours-Sup-1call`, we use an MLP to transform the LLM outputs along with adaptive additive noise to increase the cardinality of outputs while preserving or improving performance. Here, we experiment with different model configurations related to the proposed method and empirically show that the proposed method performs the best both in terms of PR-AUC and operational granularity. Let $\hat{y}^{\text{vrb}}$ be the verbalized probability of the LLM for a given input. This corresponds to the baseline method `Prompt-Naive`. Then, the prediction in `Ours-Sup-1call` is obtained as

$$\hat{y} = \sigma(f(\hat{y}^{\text{vrb}}) + \frac{z}{w})$$

where $\sigma(.)$ is the sigmoid operator, $f(.)$ is an MLP, $z \sim \mathcal{N}(0,1)$ is random noise and $w$ is a learnable parameter. We optimize the MLP parameters and $w$ to minimize the cross-entropy loss on the train set and evaluate on the test set, as done in Section 4 of the main paper. All the experiments below are performed using `Anthropic Claude 3.0 Sonnet`. We consider three different settings, namely, `No-Noise`, `Noisy-Input` and `Noise-Through-MLP`.

In `No-Noise`, we do not employ a noise term but perform supervised training of the MLP on the LLM predictions alone. The output prediction is given by

$$\hat{y} = \sigma(f(\hat{y}^{\text{vrb}}) + \frac{1}{w})$$

We retain the adaptive bias term $\frac{1}{w}$ to have comparable model complexity to the proposed configuration.

`Noisy-Input` uses the same configuration as `No-Noise` but with 'noisy' inputs. Random noise is added to the verbalized probabilities to increase their cardinality and these noisy values are then used in optimization. The output is given by

$$\hat{y} = \sigma(f(\hat{y}^{\text{vrb}} + z) + \frac{1}{w})$$

where $z \sim \mathcal{N}(0, 0.001)$.

| Approach | Prompt Summary | #LLM Calls | Aggregator | Temperature |
|---|---|---|---|---|
| p | Return prob. of input belonging to each class. | 1 | - | 0 |
| p-temp | | 20 | mean | 1 |
| p-noise | | 1 | model | 0 |
| s-$x$ | Score input class membership strength, 1 to $x$ (highest), for each class. | 1 | - | 0 |
| p-bias | We think the answer is $c_i$. Return prob. of input belonging to each class. | #classes | mean | 0 |

Table 6: Simple baselines we explore for the case study. Note that `p-temp` and `p-bias` aggregate multiple LLM predictions per sample into one, while `p-noise` aggregates one LLM prediction per sample with $z$ drawn from $\mathcal{N}(0,1)$.

`Noise-Through-MLP` uses noise directly as input to the MLP instead of as an additive term. The transformation is given by:

$$\hat{y} = \sigma(f(\hat{y}^{\text{vrb}}, z) + \frac{1}{w})$$

Figure 9 compares the precision-recall curves of the different configurations. Since `No-Noise` does not use a noise term, there is no change in the cardinality of the outputs. The corresponding 'noisy' input version `Noisy-Input` does increase cardinality but degrades performance compared to the proposed approach. Thus, we find that the noise term in `Ours-Sup-1call` is necessary for increasing cardinality while also improving performance and that its role is not limited to enabling easier optimization. The alternative configuration with noise as an additional feature also improves cardinality and achieves similar performance as proposed but has a huge gap in the operation curve, resulting in high granularity along the recall axis.

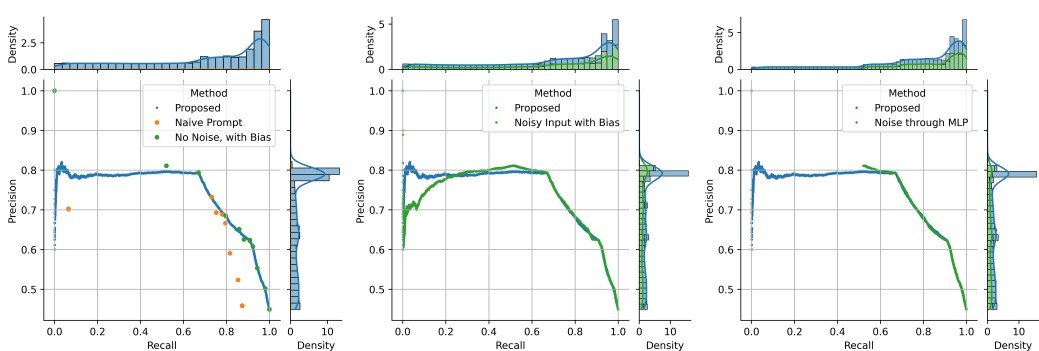

Figure 9 (a): Only MLP, no noise   Figure 9 (b): Fixed additive noise to input   Figure 9 (c): Noise through MLP

Figure 9: Ablation on proposed approach with different model configurations. We experiment with different ways to incorporate noise to increase the cardinality of the predictions. (a) We find that just supervised training without using noise helps improve the performance but cannot increase the cardinality. (b) Adding a fixed amount of noise to the original LLM predictions to first increase cardinality and then supervised training atop them results in a drop in performance. (c) Using noise as an additional feature in the MLP inputs does help increase cardinality and improve performance but does not effectively cover the precision-recall curve, resulting in a high granularity along the recall axis.

### A.8   CASE STUDY WITH ADDITIONAL PROMPTING AND UNCERTAINTY ESTIMATION BASELINES

In this section, we explore simple approaches to increase a black-box LLM's output cardinality and operational granularity. This is similar in spirit to our experiments in Section A.4 but with slightly different experimental settings and prompts. Here, we limit the discussion to SST2 dataset Socher et al. (2013) and use Claude 3.0 Sonnet Anthropic (2024) LLM for ease of experimentation. We

evaluate output distributions of different prompting approaches by investigating their ROC and PR curves, as well as their operational granularity and output cardinalities. We limit our exploration to generalizable and scalable ways of increasing the output diversity, since we aim to populate the whole operational curve as densely as possible. For this reason, we do not experiment with approaches such as providing multiple choice questions that increases the prompt length linearly in the number of unique model predictions, or asking the LLM apply a predefined function to the input. Instead, we investigate 3 main directions to enrich output distributions: (1) changing the range of outputs requested from LLM, (2) collecting and aggregating multiple responses per sample, and (3) incorporating random noise to diversify LLM predictions. For all of the approaches, we parse the LLM-decoded string output that contains numerical probabilities into floats (e.g., `"Answer: 0.61"` → 0.61).

Table 6 summarizes the different approaches we explore. The variant `p` is arguably the most straightforward way of extracting decisions in a continuous domain from black-box LLMs: by directly asking for class probabilities. This is equivalent to our baseline prompting approach `Prompt-Naive`. On the other hand, the variant `s-x` asks the LLM to return scores within given ranges. For this variant, we experiment with $x \in \{10, 100, 1000\}$. The variant `p-bias` is an uncertainty estimation-inspired approach that originates from the following observation: a bias towards one of the classes in the dataset can be provided to LLM within the prompt to impact its output. Wagner et al. (2024) observe that LLM predictions tend to vary if the uncertainty of this prediction is high, and developed an aggregation method to measure this uncertainty. Here, we use a similar approach but aggregate the predictions by taking their arithmetic mean for each class and normalize the predictions for class probabilities to sum to one. Similar to `Sample-Prob`, the variant `p-temp` extends `p` by generating multiple responses with the temperature parameter that controls the decoding stochasticity set to 1 to generate varying outputs, and aggregating them by taking their arithmetic mean as well. This is also a common approach in uncertainty estimation literature where the entropy of the response distribution per sample is a measure of uncertainty. In our case, we expect the randomness introduced by high temperature to increase the LLM output diversity. For this variant, following the author's practice, we report results by collecting 20 responses per input instance. In addition to training-free approaches, we also explore directly incorporating random noise into the predictions using a classifier, as done in the proposed approach `Ours-Sup-1call`. With `p-noise`, we utilize 10% of the available training data and learn a function $f : (z \sim \mathcal{N}(0,1), \hat{y})$ to estimate $\bar{y} = p(y|\hat{y}, z)$ to form a new operational curve. With the addition of $z$, we increase the cardinality of the inputs to $f$, while the supervised training helps retain the correctness of the output distribution. We set $f$ to a single hidden layer MLP and report mean and 95% confidence intervals of the metrics over 5 random seeds.

Table 7 summarizes the case study experiment results on the SST2 dataset. We observe that compared to directly asking the LLM to output probabilities (i.e., `p`), changing the expected output range to $[1, 10]$, $[1, 100]$ or $[1, 1000]$ does not result in consistent improvement in granularity or output cardinality while varying the performance. With `p-bias`, we observe $> 1\%$ AUC lift and more than double the output cardinality of `p`. However, this increased cardinality does not translate to a large improvement in operational granularity. `P-temp` achieves $> 20\times$ cardinality and granularity improvement with up to 1.5% AUC lift. This is because `p-temp` predictions are generated by aggregating 20 high-temperature LLM responses per input sample. Also, we observe that `p-noise` increases the output cardinality and granularity on $pr$ and $fpr$ axes further, while sacrificing performance and granularity along $tpr$. This is because `p-noise` uses $z \sim \mathcal{N}(0,1)$ as input, and it does not have a mechanism to consider the tradeoff between incorporating noise and minimizing loss. Figure 10 depicts the PR and ROC curves with their density plots of the two most promising approaches compared against the baseline `p`. To visualize densities, we estimate the probability density function (PDF) of $\Pi_a(S)$ using a Gaussian kernel: $\left\{ \forall s, \ s \in \mathbb{Z} \cap [1, \zeta], \ p = 1/s : \frac{1}{nh} \sum_{i=1}^{|\Pi_a(S)|} \frac{1}{\sqrt{2\pi}} \exp\left(-\frac{(p-p_i)^2}{2h^2}\right) \right\}$, where $h$ denotes the bandwidth size estimated using Scott's rule Scott (2015), and $\zeta$ controls the number of points the estimation is conducted at. The figure confirms that `p` has very few options to operate at compared to the other methods and depicts that the hierarchy of operational granularity and densities are consistent with each other.

We conclude that prompting does not provide comparable operational granularity other approaches, and `p-noise` is an efficient approach to improve granularity. However, it requires explicit control over the noise-performance tradeoff. We also observe that `p-bias` and `p-temp` are able to surface

| Approach | PR | | ROC | | Out Cardinality ↑ |
|---|---|---|---|---|---|
| | AUC ↑ | Granularity ↓ | AUC ↑ | Granularity ↓ | |
| p | 0.973 | 0.22, 0.71 | 0.981 | 0.61, 0.71 | 11 |
| s-10 | 0.981 | 0.33, 0.43 | 0.978 | 0.78, 0.43 | 10 |
| s-100 | 0.982 | 0.20, 0.61 | 0.985 | 0.53, 0.61 | 13 |
| s-1000 | 0.977 | 0.2, 0.61 | 0.984 | 0.53, 0.61 | 11 |
| p-bias | 0.986 | 0.19, 0.33 | 0.985 | 0.48, 0.33 | 27 |
| p-temp | **0.988** | **0.02, 0.06** | **0.989** | 0.05, **0.06** | 238 |
| p-noise | $0.975 \pm 0.01$ | $\mathbf{0.02 \pm 0.01}, 0.14 \pm 0.12$ | $0.968 \pm 0.01$ | $\mathbf{0.02 \pm 0.01}, 0.14 \pm 0.12$ | $\mathbf{480 \pm 34}$ |

Table 7: Case study experiment results with 7 baselines. Notably, output cardinality and operational granularity do not always change proportionally. This is because some output distributions result in densely populated sub-regions in PR and ROC curves.

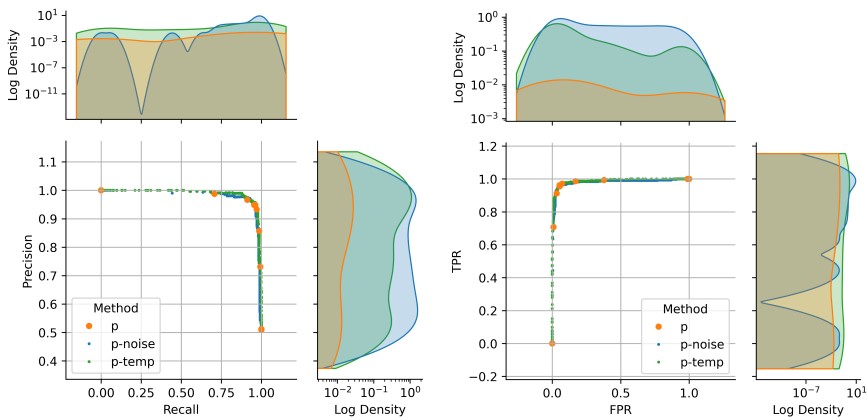

Figure 10: PR (top) and ROC (bottom) curves of `p`, `p-temp` and `p-noise` (MLP), where y-axes of density plots are in logarithmic scale for visibility.

additional information on $y$, that can potentially be utilized to reduce the negative impact of noise to performance, however, directly using them during inference is costly as they require multiple LLM calls per sample. These observations motivate our proposed methods.

A.9  EXTENDED EXPERIMENT RESULTS WITH THE PROPOSED APPROACHES

In this section, we provide additional measurements that correspond to the experiment results presented in the main body. Figure 11 extends Figure 4 that reported PRAUCs with $2\sigma$ error bars for the individual dataset (11ds) and joint dataset (joint) settings by providing AUROCs. Figures 12 and 13 show PR and ROC curves across 5 seeds, extending the Figure 5 (a): from the main body that focused on precision-recall curves of the benchmark and the proposed methods using Claude on a single seed. The optimization in the learning based approach `Ours-Sup-1call` can sometimes fail due to an incorrect choice of learning rate combined with extremely small dataset sizes (refer Figure 12(i)). We ignore such runs (and corresponding runs from other methods) when calculating the aggregated metrics. Finally, the Figures 14, 15, 16 and 17 breaks down the analysis provided for dataset-specific training and testing covered in Figure 4 (**Top**) into performance metrics on individual datasets, providing PR and ROC AUCs.

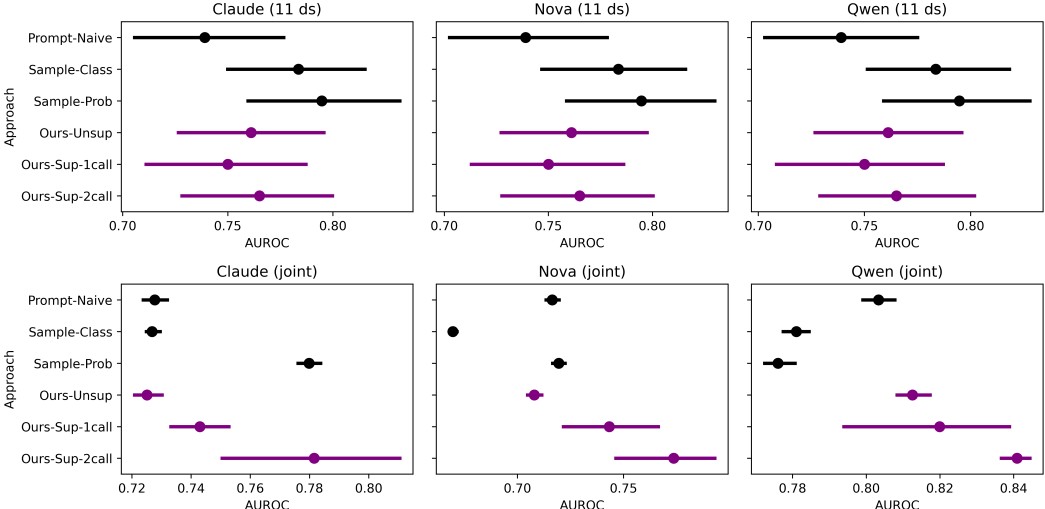

Figure 11: AUROCs of different approaches over 5 different train/test splits with $2\sigma$ error bars. Proposed methods are shown in purple. **(Top)** Aggregated results over 11 datasets. **(Bottom)** Results on combined dataset.

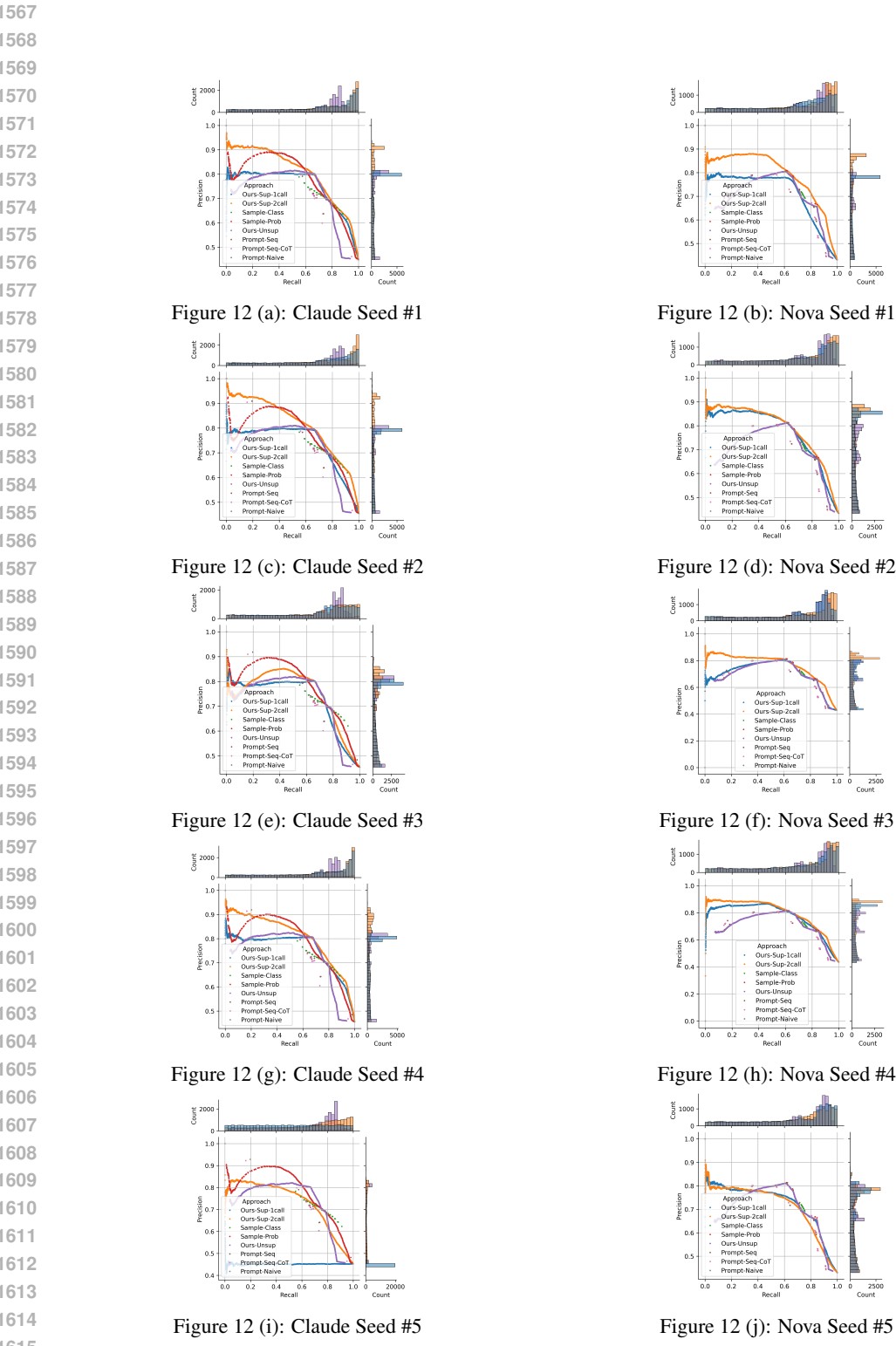

Figure 12 (a): Claude Seed #1

Figure 12 (b): Nova Seed #1

Figure 12 (c): Claude Seed #2

Figure 12 (d): Nova Seed #2

Figure 12 (e): Claude Seed #3

Figure 12 (f): Nova Seed #3

Figure 12 (g): Claude Seed #4

Figure 12 (h): Nova Seed #4

Figure 12 (i): Claude Seed #5

Figure 12 (j): Nova Seed #5

Figure 12: PR curves of the black-box LLM-based results across five different seeds.

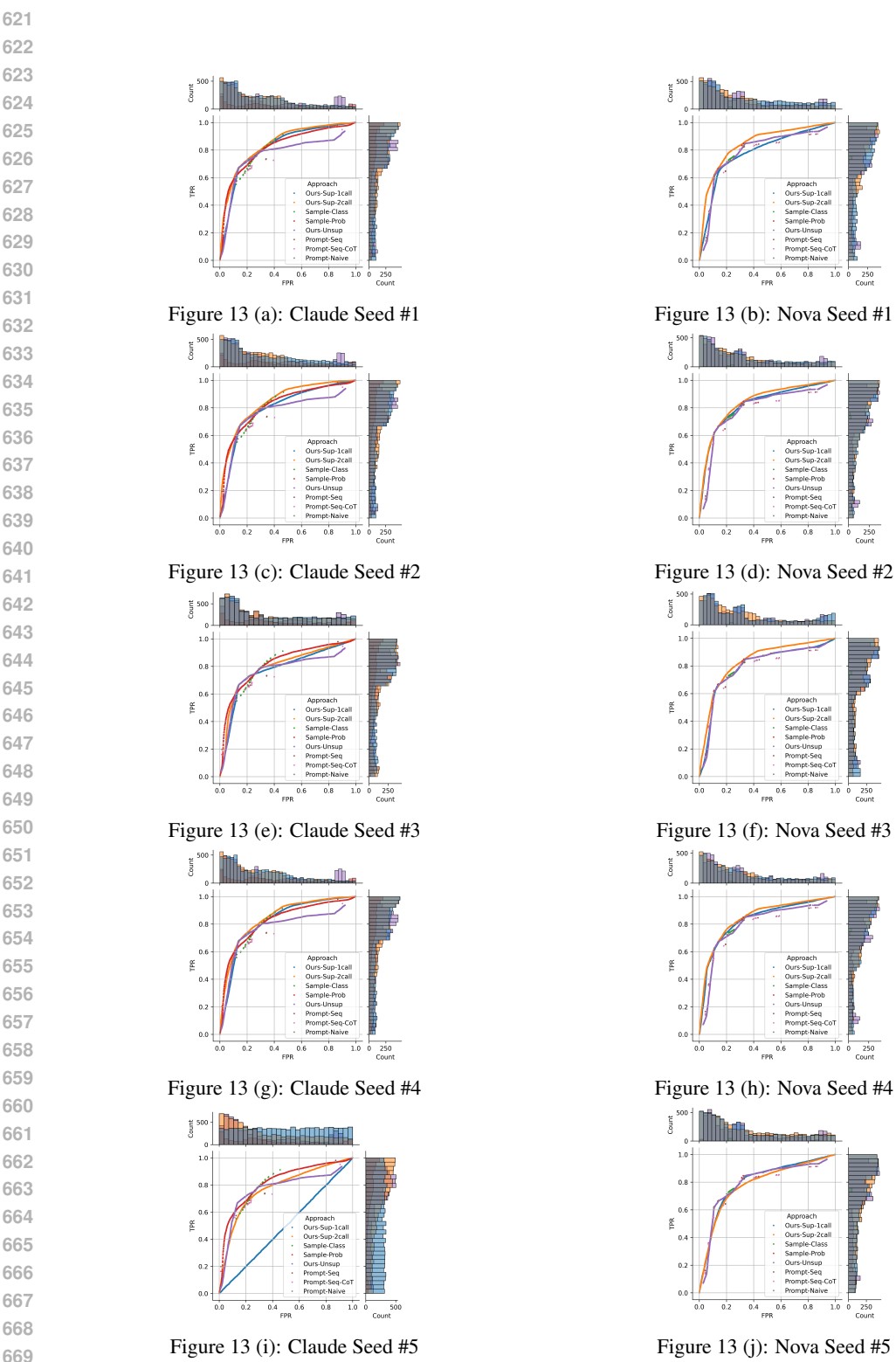

Figure 13 (a): Claude Seed #1

Figure 13 (b): Nova Seed #1

Figure 13 (c): Claude Seed #2

Figure 13 (d): Nova Seed #2

Figure 13 (e): Claude Seed #3

Figure 13 (f): Nova Seed #3

Figure 13 (g): Claude Seed #4

Figure 13 (h): Nova Seed #4

Figure 13 (i): Claude Seed #5

Figure 13 (j): Nova Seed #5

Figure 13: ROC curves of the black-box LLM-based results across five different seeds.

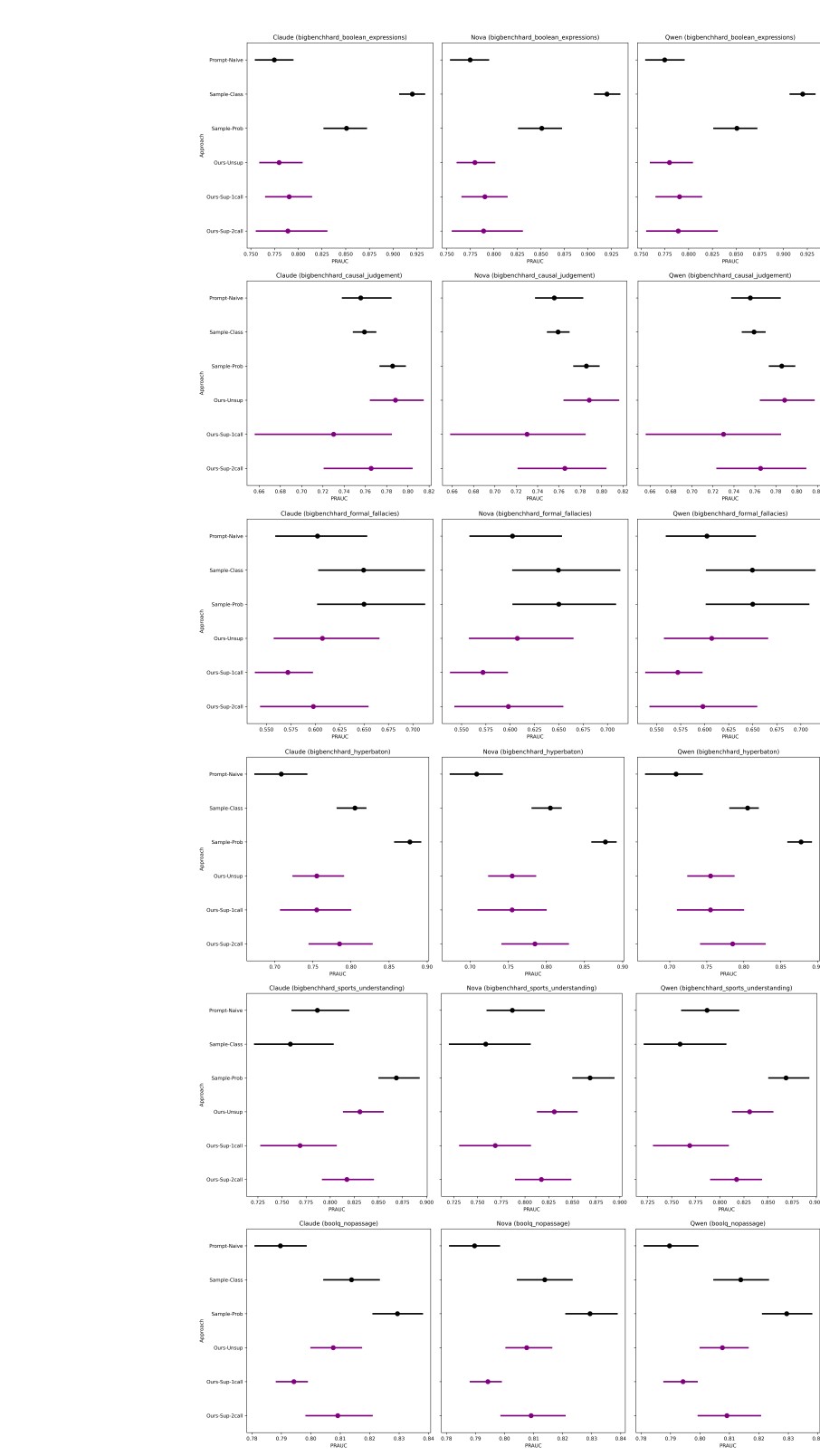

Figure 14: Dataset-level PR (part 1)

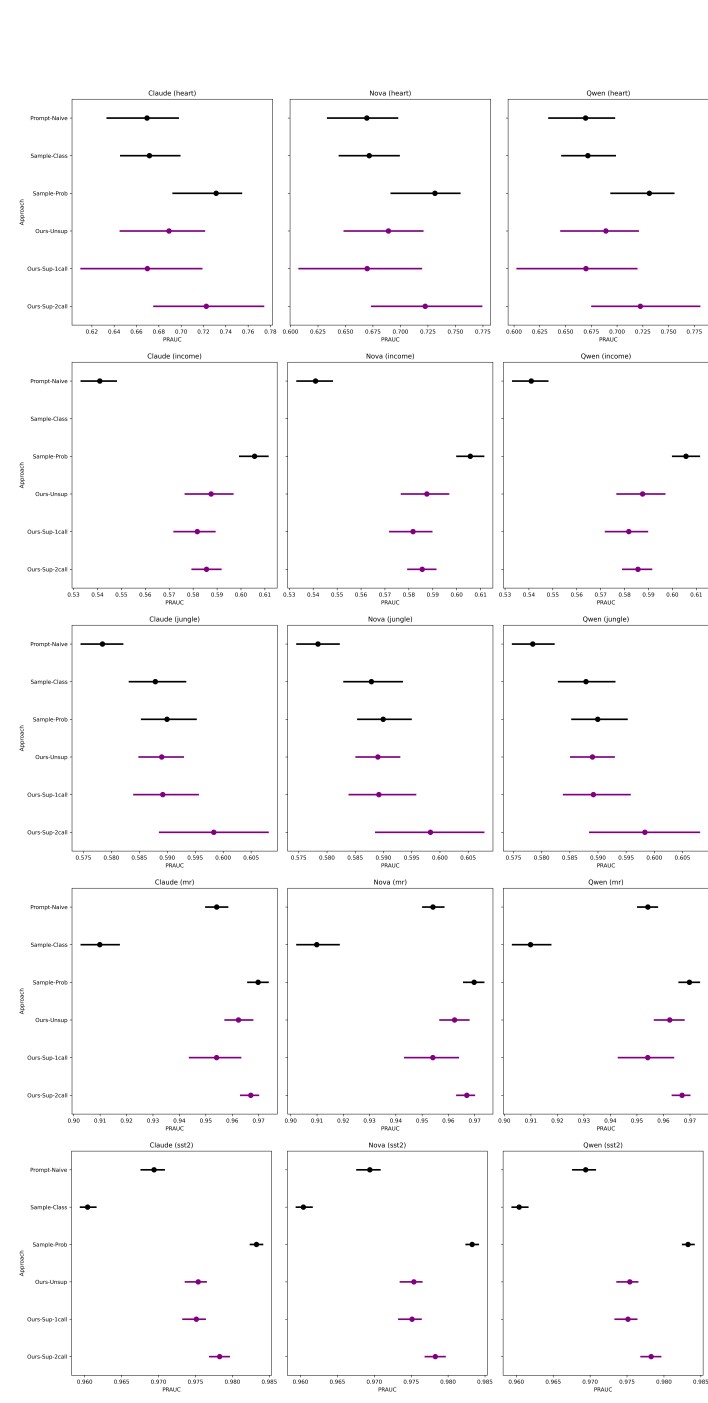

Figure 15: Dataset-level PR (part 2)

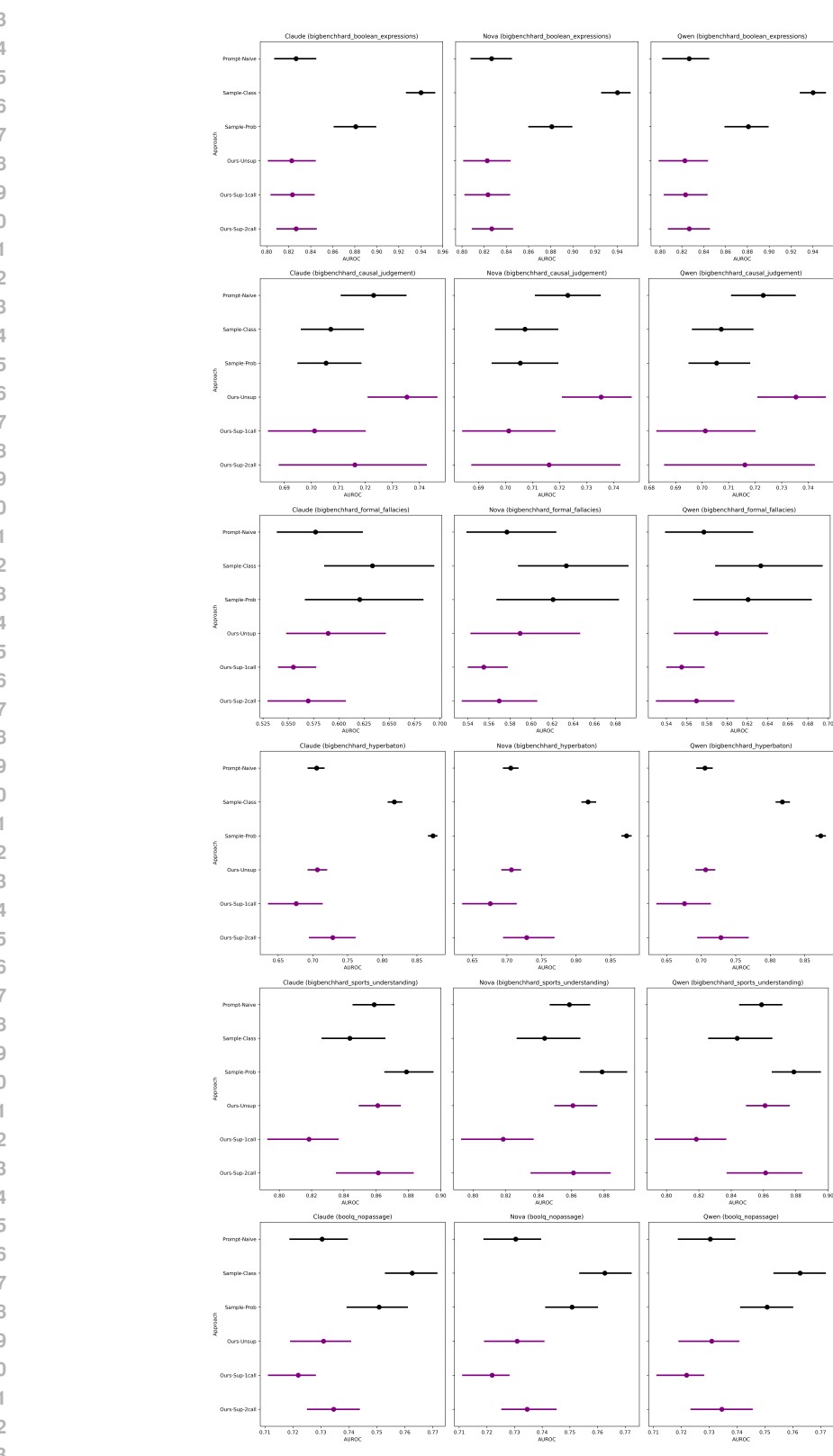

Figure 16: Dataset-level ROC (part 1)

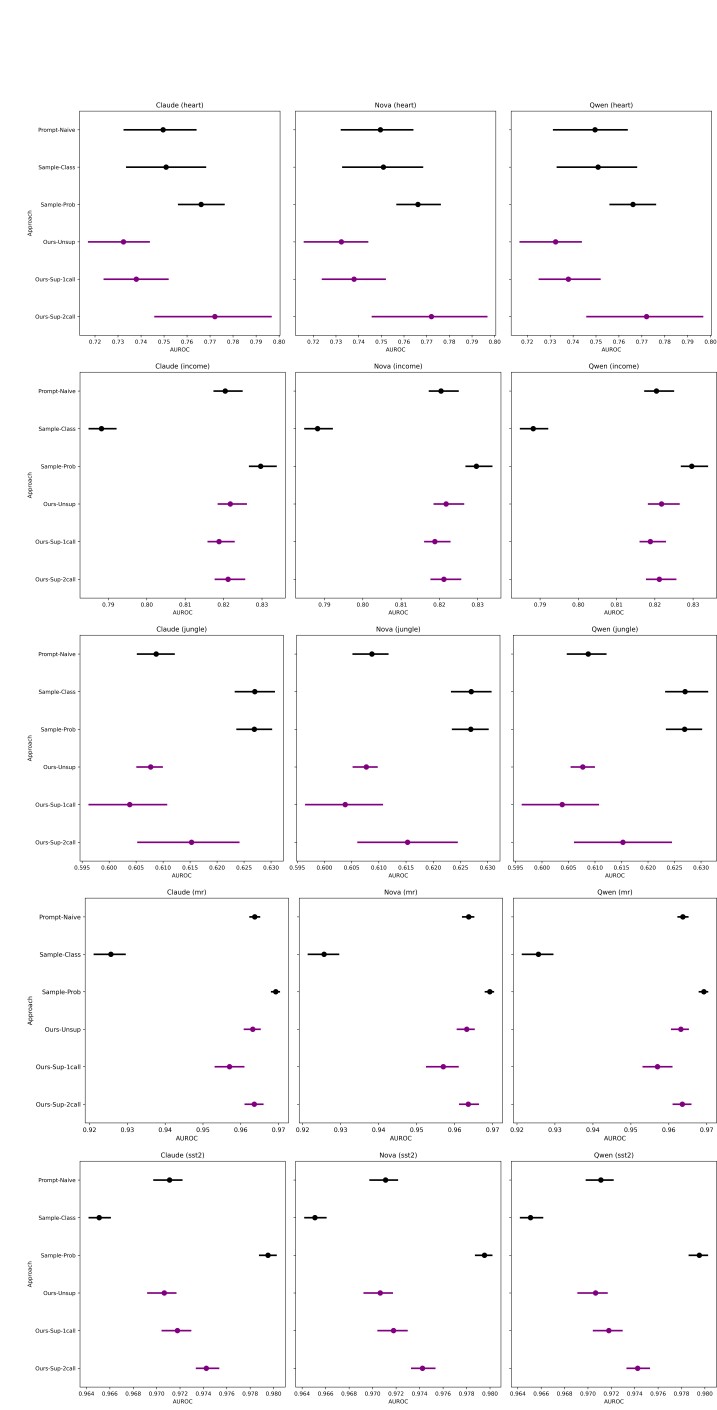

Figure 17: Dataset-level ROC (part 2)

