# OpenReview forum: "Enabling Fine-Grained Operating Points for Black-Box LLMs"
_ICLR.cc/2026/Conference — ICLR 2026 Conference Withdrawn Submission_

### Official Review · Reviewer_64ZF · 2025-10-19

**Soundness:** 2
**Presentation:** 2
**Contribution:** 2
**Rating:** 4
**Confidence:** 3

**Summary:**

The paper is organized as follows:
- The authors first conduct exploratory analysis to show that low-cardinality confidence can be observed across multiple open-weight and API-access models (e.g. Claude 3, Nova 1, and Qwen), which can result in coarse-grained operating points on PR and ROC curves.
- The authors then analyze why this "rounding bias" exist and demonstrate standard prompting techniques falter to address this problem.
- To address this challenge, the authors propose to add calibrated noise to increase output cardinality while maintaining performance. Empirical evaluations show that their methods outperform established prompting baselines.

**Strengths:**

- Enabling fine-grained control of operating points for black-box LLMs is an important desideratum, and has meaningful ramifications in high-stake decision-making scenarios such as medical treatment.
- The paper is mostly well-written, and the authors have gone lengths in providing certain definitions (e.g. PR and ROC). The (extended) literature review is also well-done with many useful references.
- The logical flow of the paper makes sense (i.e. EDA / identifying problems -> presenting hypothesis -> proposing a solution).
- The authors conduct extensive experiments and baselines (several prompting strategies) over a broad suite of binary classification tasks, and show that their proposed approach can improve granularity without sacrificing model performance on these tasks.
- The authors are honest in reporting potential limitations and the trade-off between performance and granularity.

**Weaknesses:**

- The datasets studied in this paper are sourced from well-established benchmarks (e.g. SST-2, BoolQ) which may be contained in the tested models' training set. This may lead to qualitatively different analyses compared to the high-stake decision-making scenarios that the authors are targeting (e.g. medical treatment). In the paper's experiments, models can be confident in their verbalized probabilities compared to the true zero-shot, black-box access scenario that the authors are targeting. While the reviewer acknowledges and expects the rounding bias would still persist, it would be beneficial for the authors to synthesize some simple, non-contaminated datasets to evaluate their approach.
- The proposed approach (i.e. injecting continuous noise and learning a small post-processor) is practical, but conceptually straightforward and resembles standard calibration procedures.. The authors should more sharply differentiate themselves from prior black-box calibration / uncertainty estimation work and emphasize where its contribution is methodologically new versus an engineering solution for a practical measurement gap.
- The authors sometime conflate cardinality with operational granularity (e.g. Eqn 2) but in Table 7 they are presented as separate measurements.
- Certain figures (e.g. Figure 1 and 2) contain overlapping histograms that can be a bit difficult to parse.
- (Minor point that the does not cause score decrement) While the rounding bias hypothesis is interesting, it remains a hypothesis. As far as this reviewer is aware, there are several prior studies that aim to provide mechanistic analyses of the preference to the multiples of 5 and 10, e.g. [1], and it would be great for the authors to interface with these.

[1] https://arxiv.org/abs/2406.03445

**Questions:**

NA. See weaknesses.

---

### Official Review · Reviewer_Lijg · 2025-10-28

**Soundness:** 2
**Presentation:** 3
**Contribution:** 2
**Rating:** 4
**Confidence:** 3

**Summary:**

This paper investigates how to make black-box large language models (LLMs) more controllable when used as classifiers. The authors observe that such models produce low-cardinality confidence outputs—only a few distinct probability values—making it difficult to adjust decision thresholds to meet specific performance targets (for example, precision ≥ 95%). They formalize this limitation as **low operational granularity**, meaning limited ability to fine-tune a model’s operating point on metrics such as precision or recall.

Importantly, the paper’s focus is **not** on improving model calibration or task-specific accuracy, but on increasing the granularity of the model’s decision behavior. The authors analyze why LLMs generate coarse, rounded probabilities, show that standard prompting and calibration methods do not solve this issue, and propose efficient post-processing techniques to transform these coarse outputs into finer-grained prediction distributions. Experiments on 11 datasets and 3 LLMs show that the proposed methods yield smoother and more flexible control without loss of predictive performance.

**Strengths:**

The paper provides a clear and formal definition of Operational Granularity. Under the assumption that $\hat{y}_{i}^{\text{vrb}}$ approximates $p(y_i = 1 \mid x_i)$, the authors present intuitive formulations and well-structured objectives for three proposed methods.

**Weaknesses:**

1. The motivation of the paper is not fully convincing. Since the work focuses on improving *operational granularity* rather than predictive performance, the authors should provide concrete examples or application scenarios where finer operational granularity is crucial — for instance, situations where small changes in decision thresholds have significant real-world impact.

2. The experiments are insufficient to support the paper’s motivation.

(a) It remains unclear whether $\hat{y}_{i}^{vrb}$

from the black-box LLM truly represents $p(y_i = 1 \mid x_i)$. Without this validation, the concept of cardinality in $\hat{y}_{i}^{\text{vrb}}$ may not have a clear probabilistic meaning.

(b) The generalizability of the proposed methods should be verified. Specifically, would a function $f(\cdot)$ trained on one dataset also improve operational granularity on another dataset? The current paper assumes (a) holds, but this assumption is questionable since the relationship between $\hat{y}_{i}^{\text{vrb}}$ and the true probability likely varies across models and datasets.

3. The paper has several citation formatting issues. For instance, in Line 35–37, the authors write:

    `This accessibility has fueled their widespread adoption in various applications such as fraud detection, product classification, and medical diagnosis Min et al. (2021); Zeng et al. (2024)`

    The correct form should be:
    `This accessibility has fueled their widespread adoption in various applications such as fraud detection, product classification, and medical diagnosis (Min et al., 2021; Zeng et al., 2024).`

    To fix this, the authors should use `\citep{}` instead of `\citet{}` for in-text citations that appear in parentheses.

**Questions:**

Q:  $\hat{y}_{i}^{vrb}$

is not necessarily an accurate estimate of $p(y_i = 1 \mid x_i)$. Has this assumption been validated through ablation studies? For example, have the authors compared $\hat{y}_{i}^{vrb}$ with

$\hat{y}_{i}^{tkn}$? If the correlation between them is weak, the main contributionx of this paper may not be well supported.

Q: Are the proposed methods merely overfitting the dataset? In other words, what happens if we replace the LLM outputs with values drawn from a uniform distribution $U$? What would be the performance of $f(U)$ on the test set? This experiment could clarify whether the learned function $f(\cdot)$ truly improves operational granularity or simply fits the data distribution.

---

### Official Review · Reviewer_7KGR · 2025-11-01

**Soundness:** 2
**Presentation:** 3
**Contribution:** 2
**Rating:** 6
**Confidence:** 4

**Summary:**

This paper investigates a fundamental limitation of black-box LLMs when used as classifiers: the low cardinality of their verbalized probability outputs. In other words, when required to verbalize their uncertainty, only a few distinct probability outputs are output. These coarse probability scores result in a sparse set of operating points on ROC and PR curves, restricting the ability to meet fine-grained operational constraints (e.g., achieving ≥95% precision).

The authors conducted some simple analyses and pointed out that this issue stems from human-like rounding biases in LLM-generated outputs. To address this, the authors propose a noise injection method with 3 variants—an unsupervised version and two supervised versions (single- and two-call)—that inject continuous noise (e.g., Gaussian) into verbalized probabilities and learn a small MLP correction function to adjust these noisy outputs while maintaining calibration and performance. The function adjusts the noisy outputs in a way that preserves or improves classification performance, trained using supervision when labels are available.

Experiments across 11 binary classification datasets and 3 commercial LLMs (Claude, Nova, Qwen) demonstrate that the proposed methods improve operational granularity—boosting the number of distinct thresholds from dozens to tens of thousands—while maintaining or improving PR AUC.

The contributions include: (1) characterizing the low-cardinality problem in LLM verbalized probabilities, and (2) proposing 3 algorithms to improve operational granularity in black-box settings.

**Strengths:**

1. The paper addresses a novel and underexplored limitation of black-box LLMs — the low cardinality of their verbalized probability outputs — and provides a systematic characterization of this phenomenon across multiple datasets and models.
2. The experimental setup is comprehensive: 11 binary classification datasets and 3 commercial LLMs (Claude, Nova, Qwen), with extensive comparisons against sampling-based uncertainty estimation and confidence elicitation baselines.
T3. he results demonstrate that the proposed supervised variants significantly increase operational granularity (10× more unique operating points) without degrading AUROC or AUPRC.

**Weaknesses:**

1. The empirical gains, especially for the upper plot of Figure 4, remain modest, with noticeable variance across splits. The improvement is clearer when aggregating data, but still with high variance.
2. The MLP-based correction module, though simple and effective, operates solely on the verbalized probabilities without leveraging any input-conditional information. A deeper integration with semantic or contextual features might strengthen the generalization argument.

**Questions:**

1. How is this control enforced at inference time? Is it deterministic given the same input, or stochastic across runs? If stochastic, how would a user reproduce an operating point reliably in production?

---

### Official Review · Reviewer_DaWY · 2025-11-01

**Soundness:** 2
**Presentation:** 2
**Contribution:** 1
**Rating:** 2
**Confidence:** 5

**Summary:**

The authors investigate the fact that LLMs’ verbalized confidence estimates emphasize round numbers. They propose methods for injecting randomness into verbalized confidence estimates, so that the post-processed confidence estimates yield smoother PR/ROC curves.

**Strengths:**

- Paper demonstrates empathy with practitioners who wish to construct PR/ROC charts from non-smooth data.

**Weaknesses:**

- It’s well known in the community that LLMs’ verbalized confidence estimates emphasize round numbers, mimicking everyday speech by humans. In my opinion, this is neither a mystery nor a surprise.
- LLMs under evaluation are somewhat dated (Claude 3? I haven’t heard this name in a long time…).
- The proposed methodology injects randomness so that verbalized confidences present the appearance of being more granular without actually being more informative. This approach doesn’t address the fundamental problem, which is that LLMs give coarse-grained uncertainty estimates. The authors mention the possibility of finetuning LLMs to make verbalized confidence estimates more fine-grained: this would be a more promising approach.
- The proposed smoothing method appears biased since it adds positive numbers to the verbalized confidence estimates (a positive weight w > 0 is multiplied by z ~ U(0,1)). I understand that the authors emphasize preserving the relative ranking of verbalized confidence estimates, but this scope seems too narrow, since obtaining calibrated uncertainty estimates is of great interest to practitioners.
- Many of the displayed results have very wide confidence intervals, making comparisons tenuous.

**Questions:**

- What exactly is the function f in Equation (5)?

See section on weaknesses.

---

### Note · Authors · 2025-12-05

I have read and agree with the venue's withdrawal policy on behalf of myself and my co-authors.